# The *C. elegans* GATA transcription factor *elt-2* mediates distinct transcriptional responses and opposite infection outcomes towards different *Bacillus thuringiensis* strains

**Alejandra Zárate-Potes**[1¤a], **Wentao Yang**[1¤b], **Barbara Pees**[1¤c], **Rebecca Schalkowski**[1¤d], **Philipp Segler**[1], **Bentje Andresen**[1], **Daniela Haase**[1], **Rania Nakad**[1], **Philip Rosenstiel**[2], **Guillaume Tetreau**[3], **Jacques-Philippe Colletier**[3], **Hinrich Schulenburg**[1,4‡]*, **Katja Dierking**[1‡]*

1 Department of Evolutionary Ecology and Genetics, Christian-Albrechts-Universität zu Kiel, Kiel, Germany, 2 Institute for Clinical Molecular Biology (IKMB), Christian-Albrechts-Universität zu Kiel, Kiel, Germany, 3 Univ. Grenoble Alpes, CNRS, CEA, Institut de Biologie Structurale, Grenoble, France, 4 Max Planck Institute for Evolutionary Biology, Ploen, Germany

¤a Current address: Division of Biomedical and Life Sciences, Lancaster University, Lancaster, United Kingdom
¤b Current address: Department of Pharmaceutical Science St. Jude Children's research Hospital, Memphis, Tennessee, United States of America
¤c Current address: Department of Integrative Biology, University of California, Berkeley, Berkeley, United States of America
¤d Current address: Department of Ecology and Evolutionary Biology, University of Toronto, Toronto, Ontario, Canada
‡ These authors are joint senior authors on this work.
* hschulenburg@zoologie.uni-kiel.de (HS); kdierking@zoologie.uni-kiel.de (KD)

## Abstract

The nematode *Caenorhabditis elegans* has been extensively used as a model for the study of innate immune responses against bacterial pathogens. While it is well established that the worm mounts distinct transcriptional responses to different bacterial species, it is still unclear in how far it can fine-tune its response to different strains of a single pathogen species, especially if the strains vary in virulence and infection dynamics. To rectify this knowledge gap, we systematically analyzed the *C. elegans* response to two strains of *Bacillus thuringiensis* (Bt), MYBt18247 (Bt247) and MYBt18679 (Bt679), which produce different pore forming toxins (PFTs) and vary in infection dynamics. We combined host transcriptomics with cytopathological characterizations and identified both a common and also a differentiated response to the two strains, the latter comprising almost 10% of the infection responsive genes. Functional genetic analyses revealed that the AP-1 component gene *jun-1* mediates the common response to both Bt strains. In contrast, the strain-specific response is mediated by the *C. elegans* GATA transcription factor ELT-2, a homolog of *Drosophila* SERPENT and vertebrate GATA4-6, and a known master regulator of intestinal responses in the nematode. *elt-2* RNAi knockdown decreased resistance to Bt679, but remarkably, increased survival on Bt247. The *elt-2* silencing-mediated increase in survival was characterized by reduced intestinal tissue damage despite a high pathogen burden and might thus involve increased tolerance. Additional functional genetic analyses confirmed the

**Data Availability Statement:** All relevant data are within the manuscript and its Supporting Information files.

**Funding:** This project was funded by the German Science Foundation https://www.dfg.de/ (DFG grant DI 1687/2-1 to KD and DFG grant SCHU 1415/15-1 to HS) and benefited from support by the Agence Nationale de la Recherche https://anr.fr/ (grants ANR-17-CE11-0018-01 and ANR-2018-CE11-0005-02 to J.-P.C.). AZ and WY were also funded by the IMPRS for Evolutionary Biology, Germany; HS by a Max-Planck Fellowship, HS and KD by institutional funds from Kiel University; PR and HS, and KD additionally by the DFG under Germany's Excellence Strategy – EXC 22167-390884018. The funders had no role in study design, data collection and analysis, decision to publish, or preparation of the manuscript.

**Competing interests:** The authors have declared that no competing interests exist.

involvement of distinct signaling pathways in the *C. elegans* defense response: the p38-MAPK pathway acts either directly with or in parallel to *elt-2* in mediating resistance to Bt679 infection but is not required for protection against Bt247. Our results further suggest that the *elt-2* silencing-mediated increase in survival on Bt247 is multifactorial, influenced by the nuclear hormone receptors NHR-99 and NHR-193, and may further involve lipid metabolism and detoxification. Our study highlights that the nematode *C. elegans* with its comparatively simple immune defense system is capable of generating a differentiated response to distinct strains of the same pathogen species. Importantly, our study provides a molecular insight into the diversity of biological processes that are influenced by a single master regulator and jointly determine host survival after pathogen infection.

## Author summary

Although invertebrates possess a comparatively simple immune system, research over the past decades revealed that a complexity of different signaling processes interact to produce fine-tuned defense responses to various pathogenic challenges. To date, however, it is still largely unexplored to what extent invertebrates generate a differentiated response against different strains of a given pathogen species. Here, we used the nematode *C. elegans* as a model to elucidate the common and, importantly, distinct defense responses directed at different strains of the same pathogen taxon, *Bacillus thuringiensis* (Bt), which vary in infection characteristics. We found that silencing of a single GATA transcriptional regulator causes protection against one Bt strain but sensitization against another. The protection is multifactorial and most likely mediated via tolerance (i.e., ability to limit cellular damage despite high pathogen burden), while sensitization results from decreased resistance (i.e., ability to limit pathogen burden through induction of antimicrobial effectors). Our work demonstrates that invertebrate defense responses against two different strains of the same pathogen species can be distinct, that they likely involve tolerance against one of the strains and are mediated by a single transcription factor as a central master switch.

## Introduction

In nature animal hosts are commonly exposed to different strains of a given pathogen species, which may vary in virulence factor expression and infection dynamics. It is however largely unknown how such variation between multiple pathogen strains of a given species affect the host's first line of defense: the innate immune response. Invertebrates are excellent models to address this question because they rely exclusively on innate immunity to fight pathogen threats. While it is established that distinct signaling pathways regulate invertebrate defense responses to broadly different groups of pathogens [1–3], phenotypic evidence suggests that invertebrate defense responses can even be distinct between different strains of the same pathogen species [4–11]. However, only few studies directly tested how different strains of a pathogen affect the host immune response. In bumblebees, transcriptome analyses revealed striking differences in the host gene expression response to different genotypes of the trypanosome gut parasite *Crithidia bombi*, which vary in infectivity [12]. In the fruit fly *Drosophila melanogaster*, different genotypes of the parasitoid wasp *Leptopilina boulardi*, which differ in virulence, differentially affect cellular and humoral immune responses [13].

Similarly, in the schistosomiasis vector snail *Biomphalaria glabrata* different strains of *Schistosoma* parasites induce common and distinct host expression responses and combinatorial activation of putative pathogen recognition molecules have been proposed to explain specificity of immunological memory in this system [9]. Yet, these findings only apply to eukaryotic parasites and have not been validated by functional genetic analysis. In sum, little is known about whether and how different strains of bacterial pathogen species induce differential invertebrate immune responses.

The nematode *Caenorhabditis elegans* is one of the main models to study the function of the innate immune system against bacterial pathogens. *C. elegans* is known to express common, but also specific transcriptional responses towards distantly related bacterial pathogen species with different virulence mechanisms and infection dynamics [14–19]. It is proposed on the one hand, that the *C. elegans* general response to pathogens includes the activation of central transcriptional regulators, such as the GATA family of transcription factors and conserved innate immune and stress response pathways [14,17]. On the other hand, the distinct responses to different pathogen species are proposed to be regulated by joint action of different transcription factors and mediated by potentially specific antimicrobial effectors [14,17,20,21]. Yet, to date, it remains unknown how distinct strains within a bacterial species differentially influence the nematode's immune response. Here we addressed this question by analyzing the *C. elegans* response to two closely related strains of *Bacillus thuringiensis* (Bt) that produce different pore forming toxins (PFTs) and vary in infection dynamics.

Bt is characterized by expression of a variety of crystal (Cry) PFTs that define to a large extent the bacterium's host specificity, including pathogenicity towards nematodes [22]. The *C. elegans* defense response to *Escherichia coli* expressing the single three-domain (3d) Cry toxin Cry5B was studied in detail previously and several signaling pathways required for defense against Cry5B were identified [23–29]. One study assessed the proteomic response of *C. elegans* to *E. coli* expressing Cry6Aa, which exhibits a distinct protein structure from Cry5B [30]. As yet, however, the molecular basis of specific interactions between *C. elegans* and distinct Bt strains in a natural infection setting has not been systematically analyzed. In this study we focused on two nematocidal Bt strains MYBt18247 (Bt247) and MYBt18679 (Bt679), which belong to the same bacterial species but vary in pathogenicity [31–33], in the Cry PFTs produced, and most likely also in the expression of additional virulence factors [31,32,34,35]. The strain Bt407 served as a non-pathogenic, non-Cry PFT-producing control [36]. The generally more virulent Bt679 produces the nematocidal 3d-Cry PFTs Cry21Aa3 (Cry21Aa) and Cry14Aa2 (Cry14Aa) [32], while the generally less virulent Bt247 only expresses a unique, structurally distinct toxin, Cry6Ba [31]. Despite the differences in structure (and potential mechanism of action) of these Cry toxins, the form of cellular damage they cause is similar in that they all penetrate the target cell membrane and generate pores, eventually leading to cell death.

We used cytopathological analyses by transmission electron microscopy (TEM) and a transcriptomic approach to assess the infection dynamics and the response of *C. elegans* to these two pathogen strains. We subsequently combined functional genetic analysis, additional transcriptomic data, comprehensive phenotyping and microscopic analyses to further characterize the common and distinct responses to Bt247 and Bt679. We found that there are, indeed, distinct responses to the two Bt strains. These are mediated by the intestinal GATA transcription factor ELT-2 and further involve resistance to infection to Bt679 and shifts in lipid metabolism, detoxification, and trehalose catabolism that putatively lead to tolerance to infection with Bt247.

## Results and discussion

### Bt247 and Bt679 vary in cytopathology and infection dynamics

We first investigated differences in infection dynamics between Bt247 and Bt679 when we infected worms with the same concentration of Bt spores. We here define infection as the accumulation of Bt spores inside the worm's gut and/or proliferation of Bt vegetative cells in the gut or throughout the whole worm body. Spore accumulation in the gut and proliferation of vegetative cells in dead worms haves been repeatedly shown for both Bt247 and Bt679 in the past [31,32,34,35] (see also below). In detail, transmission electron microscopy (TEM) of cross sections of the *C. elegans* intestine from Bt247- and Bt679-infected living worms were inspected 2 hours (h), 6 h, 12 h, and 24 h post infection (p.i.) (Fig 1A) and compared to worms exposed to non-pathogenic Bt407 or to the *C. elegans* laboratory food bacterium *E. coli* OP50 (Fig 1B–1Q). In the *E. coli*-fed control worms, all bacteria were broken up, and the worm displayed a thin intestinal lumen with normal, long and straight microvilli (Fig 1E, 1I, 1M and 1Q). *C. elegans* on the non-pathogenic Bt407 displayed slightly widened intestinal lumens and accumulation of Bt407 spores at later time points (12 h and 24 h p.i.), but also no signs of tissue damage (Fig 1D, 1H, 1L and 1P). Infection with Bt679, however, resulted in severe tissue damage as early as 6 h p.i. (Fig 1C and 1G), leading to early spore germination (12 h p.i.), accumulation of vegetative cells, and host killing (Fig 1K and 1O). In contrast, infection with Bt247 resulted in gradually increasing damage of intestinal tissue and accumulation of spores in the intestinal lumen over time, but apparently no germination within live hosts (Figs 1B, 1F, 1J, 1N, 4D and 4J). These observations are consistent with clearly distinct infection patterns (assessed by measuring infection load) caused by the two Bt strains and with Bt679 exhibiting higher pathogenicity as reported previously [35]. Together, our results show that dynamics of Bt247 and Bt679 infection are qualitatively different.

### The *C. elegans* transcriptomic response to Bt247 and Bt679 infection has common and distinct signatures

To understand how *C. elegans* responds to two pathogenic strains of the same species that vary in virulence factor expression and exhibit distinct infection dynamics, we explored the worm's transcriptional response to Bt247 and Bt679 across four time points p.i. by mRNA sequencing (Fig 2A). We identified 5477 genes with significant differential expression (DE) between pathogenic Bt and the non-pathogenic control strain Bt407 treatments for at least one of the time points (S1 Table). Importantly, the two control treatments Bt407 and *E. coli* OP50 showed only little variation to each other (S1 Table), indicating that virulence-associated processes are required to activate the *C. elegans* response to Bt. We thus focused on the comparison between the pathogenic Bt strains (Bt247 and Bt678) and the non-pathogenic Bt407 (Fig 2A). Using K-means clustering, we found ten distinct groups of genes that exhibited consistent differential expression patterns (Fig 2B). Despite the different Cry toxins produced by Bt247 and Bt679 and despite the clearly distinct infection dynamics, the vast majority of differentially regulated genes (91%) were common between the two strains across the four time points. Clusters 1, 3–7, 9, and 10 include genes either up-regulated (46% of all DE genes) or down-regulated (45% of all DE genes) by infection with both pathogenic strains and thus reflect the common transcriptional response to Bt (Fig 2B, S1 Table).

Importantly, 9% of DE genes are found in cluster 2 (234 genes) and cluster 8 (275 genes) and these clusters define response modules that are distinct between the two Bt strains. Cluster 2 contains genes that are exclusively up-regulated upon Bt247 exposure. The potential effector genes *lec-6*, known to be required for the response against the PFT Cry6Aa [30] and the

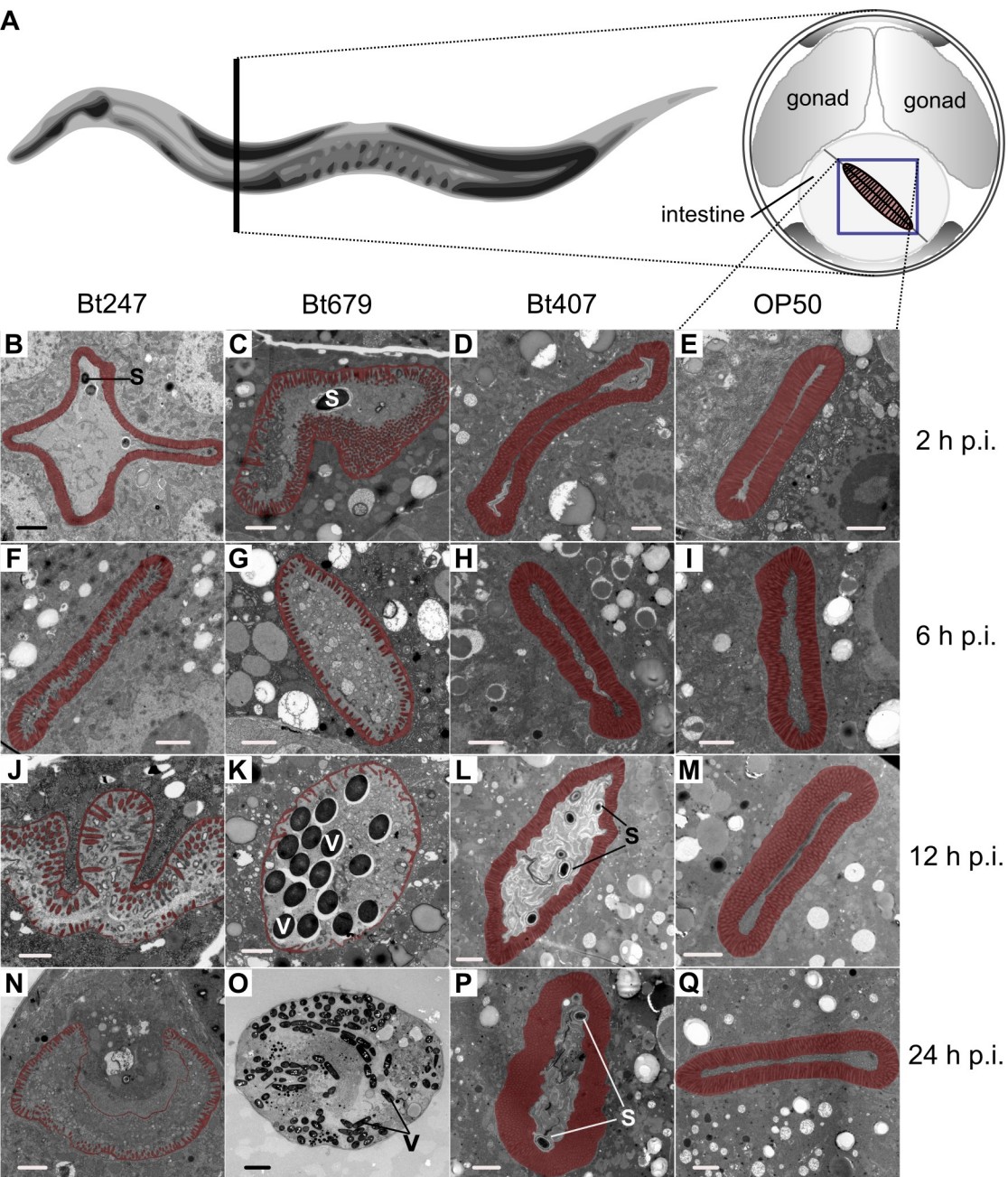

**Fig 1. Distinct infection dynamics of Bt247 and Bt679 infection.** TEM pictures of transversal cuts of the *C. elegans* intestine 2 hours (h), 6 h, 12 h, and 24 h post infection (p.i.) with *Bacillus thuringiensis* (Bt) Bt247, Bt679 or the non-pathogenic controls Bt407 and *E. coli* OP50. (A) Worm diagram modified from WormAtlas [37] showing the area of the intestine analyzed by transmission electron microscopy (TEM) for (C-Q). (B) shows the first intestinal ring. (B-Q). Microvilli of the apical membrane are marked throughout and artificially colored in red to point out integrity during infection. S = spores, V = vegetative Bt cells. Scale bar represents 1 μm except for pictures B where it is 2 μm and O where it is 5 μm. The Section delimited by a blue square in panel (A) indicates area shown in pictures (B-N, P-Q).

lysozyme *lys-2*, important for resistance against Bt247 [38], were found in this cluster. Cluster 8 contains genes, whose expression was down-regulated at early time points on both Bt247 and Bt679 but up-regulated only upon exposure to Bt247 at later time points. In this cluster,

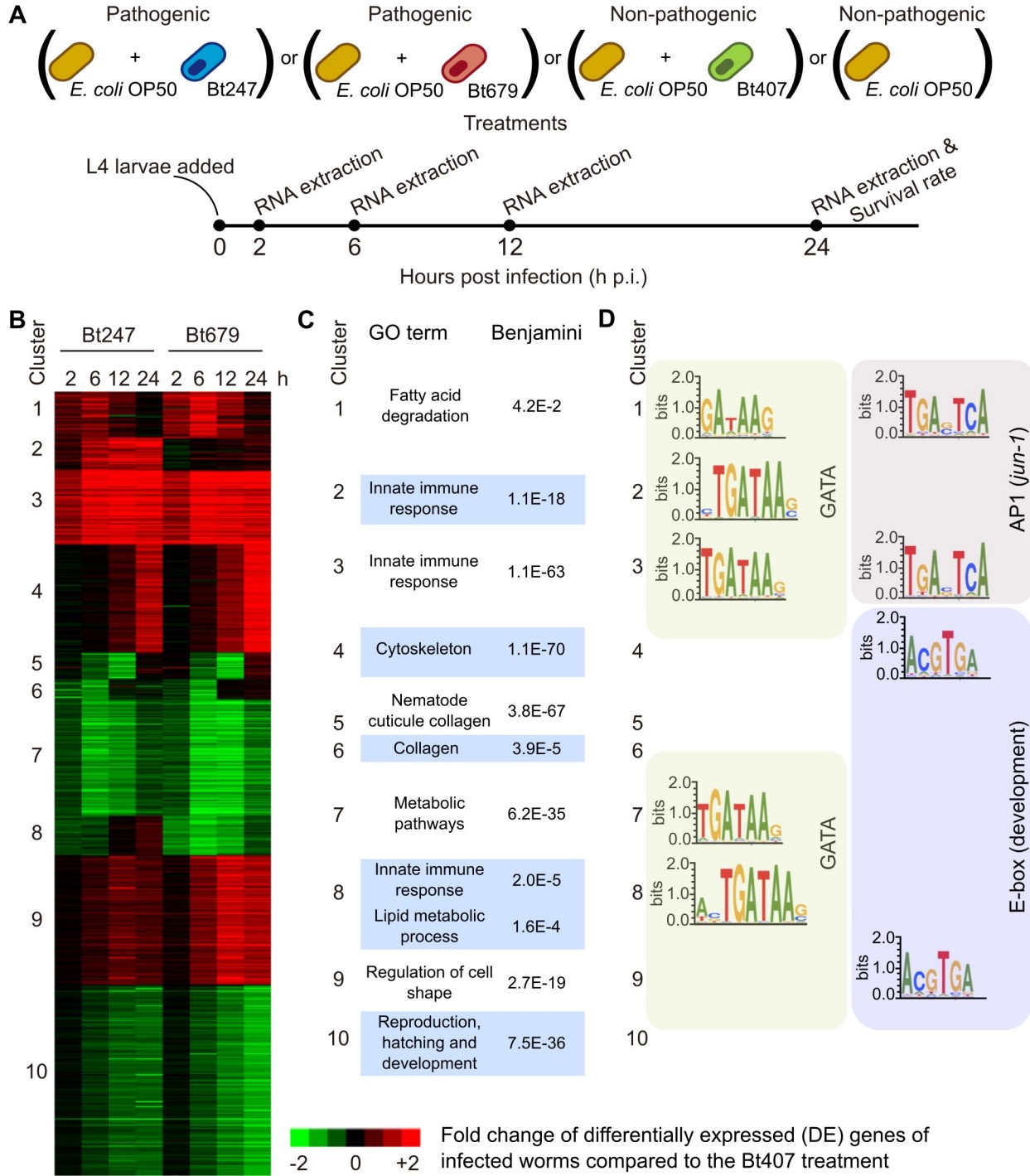

**Fig 2. Bt247 and Bt679 induce common as well as distinct transcriptomic responses in *C. elegans*.** (A) Infection protocol for the transcriptome analysis. (B) Heatmap representing all significant DE genes upon exposure to either Bt247 or Bt679 compared to the non-pathogenic Bt407 control. K-means clustering yielded ten clusters of co-expressed genes. (C) Top scores of functional enrichment analysis per cluster (p-values are Benjamini corrected). (D) Significantly enriched TF binding motifs.

we found the necrosis terminal aspartic protease gene *asp-1*, which mediates susceptibility to the PFT Cry6Aa [39], and the saposin-like protein encoding gene *spp-8*, a putative immune effector (Fig 2B, S1 Table). Genes with immunity and lipid metabolism annotation terms were significantly enriched in these two strain-specific clusters (Fig 2C, S2 Table).

To identify transcription factor (TF) binding sites that govern the common and specific gene expression changes upon Bt infection, we used a motif enrichment analysis of promoter regions for each of the gene expression clusters. A GATA motif was significantly overrepresented in the promoter regions of genes from five clusters (clusters 1–3, 7, 8), thus including the distinct response modules (Fig 2D). The binding motif of the AP-1 TF complex (JUN-1/FOS-1) was additionally enriched in clusters 1 and 3 (Fig 2D), containing genes commonly up-regulated early (Fig 2B). GATA TFs have been described as central regulators of general defense responses against pathogens in the *C. elegans* intestine [17,40–42]. The *C. elegans* JNK-like MAPK pathway, which can activate AP-1-dependent transcription, is known to regulate defenses against the Bt crystal PFT Cry5B (not expressed by either Bt247 or Bt679) as well as abiotic stresses like cadmium [27]. We conclude that GATA TFs and the JNK-like MAPK pathway represent interesting candidates for regulating distinct and common responses to strains Bt247 and Bt679, respectively.

### *jun-1* mediates a common, while *elt-2* mediates a distinct response to Bt247 and Bt679

To further characterize the common and distinct response to Bt247 and Bt679 infection, we initially tested if the AP-1(JUN-1/FOS-1) TF complex is required for resistance to both Bt strains. The survival of *jun-1(gk551)* knockout mutants was assessed upon exposure to different Bt247 and Bt679 spore dilutions at 24 h p.i. (Fig 3A and 3B). Here and below, our survival data are presented as survival curves (Fig 3A and 3D), but also as heatmaps (Fig 3E), to facilitate the comparison of results between different knockout mutants or RNAi treatments and highlight variation across technical and biological replicates. We found that *jun-1(gk551)* mutants are highly susceptible to both Bt strains (Fig 3A, 3B and 3E), suggesting that JUN-1 and the AP-1 complex are required for the common *C. elegans* response to Bt and Bt-derived PFTs, extending on their known role in protecting against Cry5B [27].

As the GATA motif was enriched in the gene clusters with differential expression towards Bt247 and Bt679, we subsequently assessed whether a GATA TF could account for strain-specific responses. Three *C. elegans* zinc-finger GATA-type TFs bind to GATA motifs and are active in the adult intestine: ELT-2, ELT-4, and ELT-7 [43]. ELT-4 is believed to be non-functional [44], while ELT-2 and ELT-7 have a synergistic action in endoderm differentiation during development [45,46]. ELT-2 is a homolog of Drosophila SERPENT [47] and vertebrate GATA4-6 and likely the predominant GATA TF for intestinal gene regulation during development and also in adult *C. elegans* [17,41,43,48]. To test the involvement of the intestinal GATA TFs in the response to Bt247 and Bt679 infection, we analyzed the survival rate of *elt-7(ok835)* and *elt-7(tm840)* knockout mutant worms (S1C Fig) and of worms, in which *elt-2* expression was silenced by RNAi (*elt-2*(RNAi), efficient knockdown shown in Fig 3F). The survival assays of *elt-7(ok835)* and *elt-7(tm840)* mutants yielded inconclusive results, as the two mutant alleles affected survival in opposite ways (S1C Fig). In contrast, the results for RNAi-mediating silencing of *elt-2* were highly consistent and produced a striking phenotype: *elt-2*(RNAi) resulted in lower worm survival after Bt679 infection (Fig 3D and 3E), but significantly higher survival after Bt247 infection (Fig 3C and 3E). Importantly, *elt-2*(RNAi) did not affect worm survival on the non-pathogenic Bt407 (Fig 3E), confirming that the animals are not generally sick [40,42]. The opposite *elt-2*(RNAi) survival phenotypes after infection with Bt247 and Bt679

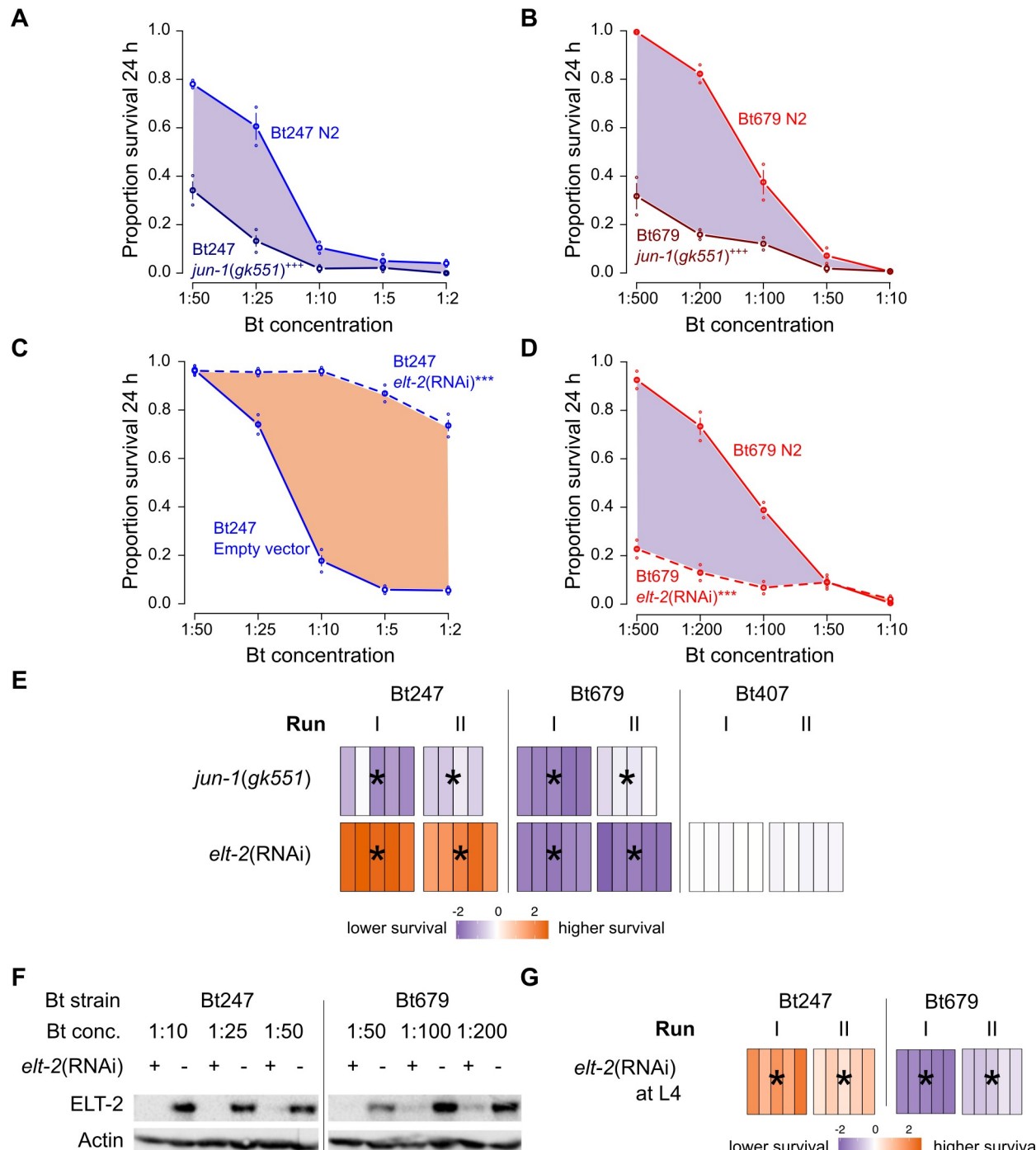

**Fig 3. *jun-1* mutants are highly susceptible to both Bt strains, while *elt-2* silencing decreased survival on Bt679 but increased survival on Bt247.** Survival of *jun-1*(*gk551*) mutant and wildtype N2 animals 24 h p.i. with (A) Bt247 and (B) Bt679. Survival of *elt-2*(RNAi) and RNAi control worms 24 h p.i. with (C) Bt247 and (D) Bt679. Each line represents proportion survival across different concentrations of Bt spores diluted in *E. coli* OP50. We chose a dose range allowing to have approximately between 95% and 5% survival from the lowest to the highest concentration of the corresponding wildtype N2 control. Control worms are represented by continuous lines in light red for Bt679 and in light blue for Bt247. The difference of area under the survival curve (AUC) is shaded in purple when the mutant/RNAi worms have lower survival rate than the controls and in orange when the mutant/RNAi worms have a higher survival rate than the controls. Mean and standard error of the mean (SEM) are shown, N = 5 plates with approximately 30 worms each. GLM of the binomial family was fitted (see supplemental methods for model), followed by a Tukey HSD Test, where mutant or knockdown worm strains were compared to control strains. Plus-signs show significant differences including all Bt concentrations between mutant and N2 (A, B), asterisks show significant differences between RNAi treatment and empty vector control (C, D). *** or +++ indicate p < 0.001, Bonferroni adjusted. Results are representative of at least two independent runs. (E) Heatmap representing the

difference of the AUC of the mutant/RNAi worms versus the average of the wildtype N2/N2 empty vector controls from the same run (i.e., biological replicate). Purple and orange colors (as in shading of A-D) indicate the value of AUC difference (see scale bar at the bottom). Bars represent technical replicates. Two runs are shown. *elt-2*(RNAi) was always performed from the L1/L2 larval stage onwards. Statistics as in A-D. Significant differences between all treatment replicates from controls are indicated by an asterisk per run. Results shown in A-D correspond to run I represented in the heat maps of E. (F) *elt-2*(RNAi) efficiently depletes ELT-2 protein levels. Western blot analysis of worm lysates from *elt-2* (RNAi) (+) and control (-) worms 6 h p.i. with Bt247 or Bt679. Upper panel, anti-ELT-2. Lower panel, anti-actin as loading control. (G) The opposite *elt-2*(RNAi) survival phenotype is not determined by *elt-2* function during development. Heatmaps show difference in survival 24 h p.i. between wildtype N2 empty vector worms and *elt-2*(RNAi) worms, with RNAi treatment starting from the L4 larval stage. Survival of three-day old adults was assessed. Data represented in heatmap and statistics as in A-D. See also S1 Fig.

were still observed when RNAi was started from the L4 stage onwards (Fig 3G) and pathogen cultures had identical virulence levels (S1B Fig compared to S1A Fig). The results are thus unlikely due to anomalous intestinal development in *elt-2*(RNAi) worms or a quantitative difference in Bt virulence.

Together, our results show that the *C. elegans* response to Bt247 and Bt679 has a common component (Figs 2B, 3A and 3B), but also a clear Bt strain-specific component evidenced by the distinct effects of *elt-2*(RNAi) on survival rate after infection (Fig 3C and 3D). While our Bt679 results support the idea of a key regulatory function of ELT-2 in *C. elegans* intestinal immune defense [17,40–42,49], the Bt247 results are unexpected as *elt-2* silencing reduces *C. elegans* susceptibility to this particular pathogen. This prompted us to further investigate the effects of *elt-2* knock-down on Bt679 and Bt247 infection processes.

## *elt-2* RNAi leads to higher intestinal tissue integrity on Bt247 and has the opposite effect on Bt679

We sought to better understand the intriguing contrasting effects of *elt-2*(RNAi) on survival of Bt247 and Bt679 infected worms and to gain insight into the underlying infection processes. For this we assessed pathogen load (Fig 4A and 4B) and intestinal integrity (Fig 4C–4N and S2) in infected *elt-2*(RNAi) and control (empty vector) worms. Previous work demonstrated that active *elt-2* can reduce bacterial load [42], presumably because it positively regulates expression of intestinal infection response genes [47]. We now found that Bt679 pathogen load of *elt-2*(RNAi) worms indeed showed a statistical trend ($0.1 > p > 0.05$) to be higher than that of the controls (Fig 4B). More importantly, however, Bt247 pathogen load of *elt-2*(RNAi) worms did not differ at all from controls ($p = 0.2$; Fig 4A). *elt-2(RNAi)* worms thus show significantly higher survival rate on Bt247 (Fig 3C and 3E), despite a high pathogen load (Fig 4A).

To assess the integrity of the intestinal epithelium of *elt-2*(RNAi) worms on Bt247 and Bt679, we performed TEM of cross sections of the anterior intestine 12 h p.i. We found that intestinal damage is generally inversely proportional to worm survival rate (Fig 4C–4N), especially for worms infected with Bt247 (Fig 4D, 4G, 4J and 4M). *elt-2*(RNAi) worms infected with Bt679 showed exacerbated loss of intestinal brush border integrity (Fig 4H–4N) compared to empty vector controls (Fig 4E and 4K). In contrast, although *elt-2*(RNAi) worms infected with Bt247 also showed damage of the intestinal epithelium (distension of the intestinal lumen, shortening of the microvilli, and gaps in the intestinal brush border; (Fig 4G and 4M), signs of damage were less pronounced than in control worms (Fig 4D and 4J). We validated and confirmed our findings by studying intestinal epithelial integrity of *elt-2*(RNAi)-treated worms in the transgenic fluorescent reporter strain BJ49, which contains the fluorescently tagged intermediate filament protein IFB-2::CFP (S2A–S2P Fig) and by using the "smurf" assay (S2Q–S2S Fig).

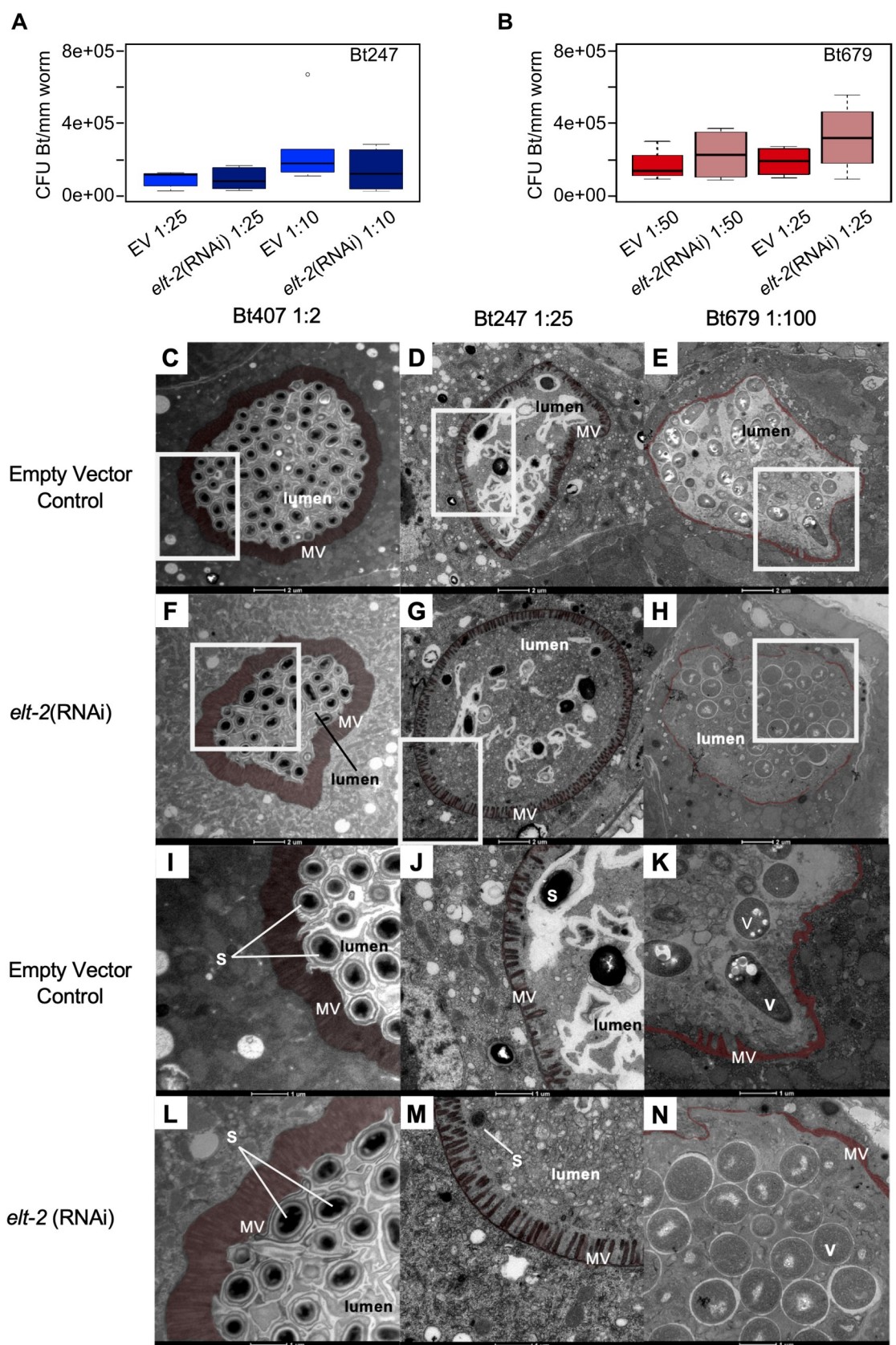

**Fig 4. *elt-2* silencing causes decreased intestinal integrity on Bt679 but increased intestinal integrity on Bt247, while pathogen load remains the same as in controls.** Bt load of *elt-2*(RNAi) and control worms exposed to (A) Bt247 and (B) Bt679. (C-N) TEM pictures of transversal sections of the anterior intestine of *elt-2*(RNAi) and control worms exposed to different Bt strains. The microvilli of the apical membrane are artificially colored in red to indicate their integrity during infection. Scale bar represents 2 μm in C-H and 1 μm in I-L. White boxes in A-F show area that is zoomed-in in I-N. MV = microvilli, S = Bt spores, V = vegetative Bt cells. See also S2 Fig.

Collectively, our results suggest that the mechanism through which *elt-2* RNAi increases host survival rate after infection with Bt247 may involve enhanced tolerance to pathogen infection, which we define as the ability to limit the damage caused by a given pathogen burden [50–53]. This response is distinct from resistance, which we define to be the ability to limit pathogen burden (following [54]), usually as a consequence of immune effector activity. *elt-2*(RNAi) worms may be tolerant to Bt247 infection, since they show a high survival rate and reduced tissue damage despite high pathogen load. On the other hand, *elt-2*(RNAi) worms are highly susceptible to Bt679 infection likely due to reduced resistance, as further discussed below.

It is tempting to speculate that the resistance vs. tolerance strategy possibly employed by the host to defend itself against Bt679 and Bt247, respectively, is advantageous in the context of the distinct infection dynamics, which we observed for the two BT strains. Since Bt679 rapidly colonizes the *C. elegans* gut (displaying early spore germination followed by massive multiplication of vegetative cells (Figs 1C, 1G, 1K, 1O, 4E and 4K)), an immediate antimicrobial response to quickly eliminate the pathogen might be the only chance of survival. Interestingly, Wang et al. observed that *C. elegans* surviving Bt679 infection show low or absent infection and proposed that an immediate activation of immune response might yield surviving hosts with cleared infections [35]. Bt247, on the other hand, displays a gradual accumulation of spores, but apparently no germination within live hosts (Figs 1B, 1F, 1J and 1N, 4D and 4J). In this case, protection against the action of the spore-associated Cry toxins by e.g. fortification of the intestinal barrier might be more advantageous for host survival than pathogen control. Indeed, an epidemiological model proposed by Restif and Koella predicts that different conditions can favor resistance or tolerance: In particular, low virulence and high infection rate should favor host tolerance, whereas high virulence and low infection rate should favor resistance [55], in general consistent with our findings.

It is striking that inactivation of a single TF gene may lead to resistance-defective animals on one Bt strain and to potentially tolerance-enhanced animals on the other Bt strain. Interestingly, in *Drosophila melanogaster*, resistance and tolerance could be increased or decreased by a single mutation in *CG3066*, a gene encoding a protease active in the melanization cascade, dependent on the pathogen species [56].

In the following sections we initiated further experiments to elucidate genes involved in *elt-2*(RNAi)-mediated susceptibility to Bt679 and high survival on Bt247.

## *elt-2* RNAi leads to reduced survival rate on Bt679 likely through reduction of resistance

We next assessed in more detail how *elt-2*(RNAi) causes reduced resistance to Bt679 infection. We first evaluated the interaction between *elt-2* and the p38 MAPK pathway, because (i) the p38 pathway is known to coordinate *C. elegans* resistance to a variety of pathogens [57] and also exposure to the Bt PFTs Cry5B and Cry21A [23,24], (ii) it cooperates with ELT-2 in the response against *Pseudomonas aeruginosa* and *Salmonella enterica* and the subsequent recovery from these infections [47,58], and (iii) the specific response clusters 2 and 8 (Fig 2B) were enriched for the GATA binding motif (Fig 2D) and also p38 pathway targets (S3 Table). We

found that the p38 MAPK mutant *pmk-1(km25)* and the p38 MAPKK mutant *sek-1(km4)* were indeed highly susceptible to Bt679 (Fig 5A). However, they surprisingly survived infection with Bt247 as well as wildtype worms (Fig 5A). *pmk-1(km25)*;*elt-2*(RNAi) animals displayed a higher survival rate compared to *elt-2*(RNAi) and control animals after Bt247 infection (Fig 5A, S3A Fig). *pmk-1(km25)*;*elt-2*(RNAi) animals were, however, significantly more susceptible than *elt-2*(RNAi) and *pmk-1(km25)* mutant worms alone after Bt679 infection, indicating that *elt-2* acts in parallel to the p38 MAPK pathway in mediating resistance to Bt679 (Fig 5A, S3B Fig), possibly by regulating the expression of complementary sets of immune effector genes. We identified two candidate Bt679-specific effector genes in our transcriptome data and tested if they play a role in the specific interaction with Bt679: The saposin-like protein encoding *spp-8* and the superoxide dismutase gene *sod-3* were indeed specifically required for resistance to

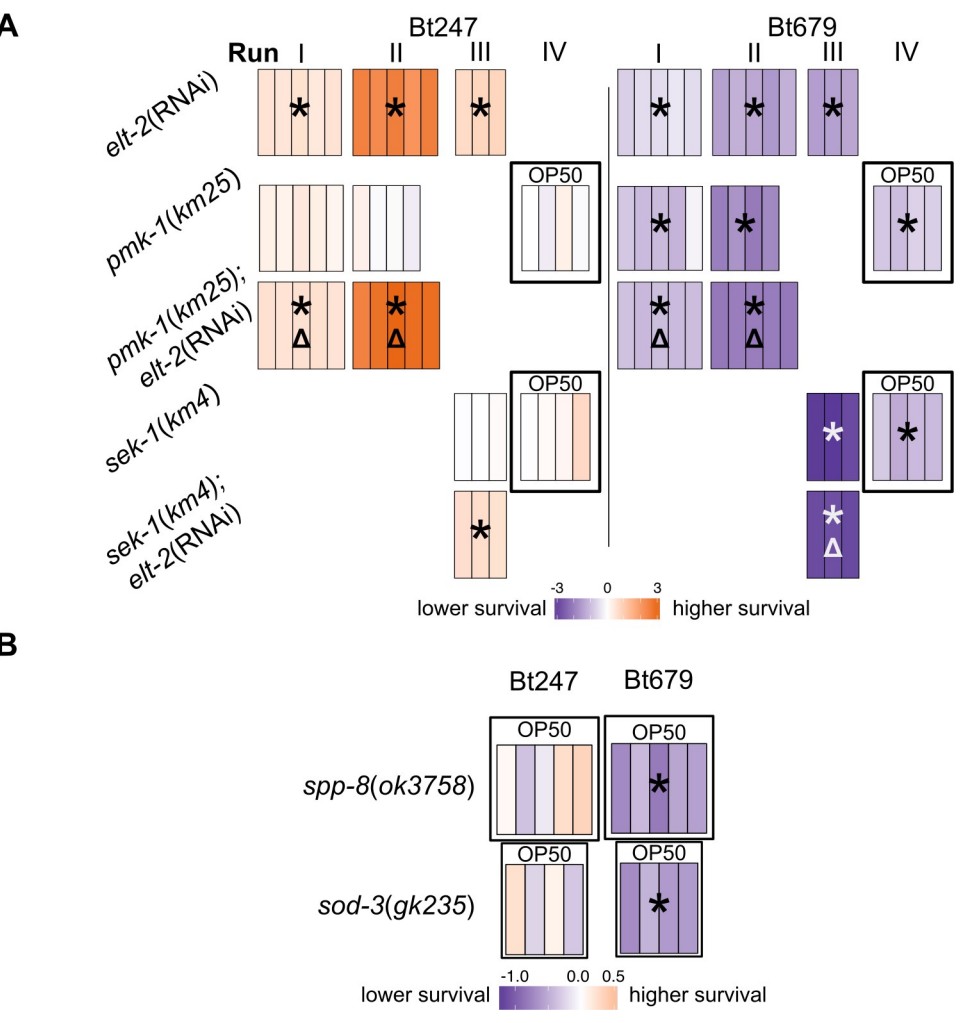

**Fig 5. The p38 MAPK pathway and putative effector genes *spp-8* and *sod-3* are specifically required for defense against Bt679 infection.** Difference in survival at 24 h p.i. between the N2 control and (A) p38 MAPK *pmk-1(km25)* and MAPKK *sek-1(km4)* mutant exposed to Bt247 or Bt679, and epistasis analysis following *elt-2*(RNAi), (B) the putative effector saposin-like protein-encoding gene mutant *spp-8(ok3758)* and the superoxide dismutase mutant *sod-3(gk235)*. *spp-8* and *sod-3* are specifically required for resistance against Bt679, but not Bt247. Data represented in heatmaps and statistics as in Fig 3E. Triangles indicate significant difference compared to *elt-2*(RNAi). In A, worms were grown on RNAi *E. coli* HT115 plates until L4 larval stage when worms were exposed to Bt. OP50 above a run indicates that the worms were grown on NGM plates seeded with *E. coli* OP50 (absence of RNAi treatment). See also S3 and S4 Figs.

Bt679, but did not affect survival on Bt247 (Fig 5B). We conclude that the p38 MAPK pathway is required for survival on Bt679 but is dispensable for defense against Bt247. These results thus confirm that part of the *C. elegans* response to Bt247 and Bt679 infection is distinct and relies on different intracellular signaling pathways.

Bt247 and Bt679, albeit being two strains of the same pathogen species, produce different Cry toxins, which are essential virulence factors of these two strains (S4E and S4H Fig) [31,32]. As *C. elegans* is capable of mounting distinct responses to these Bt strains in an *elt-2*-dependent manner, we asked if *elt-2* is also required for mounting distinct responses to individual Bt247 and Bt679 Cry toxins (S4 Fig). Cry6Ba is the only crystal protein produced by Bt247 (S4B and S4C Fig) and its depletion was shown to abolish Bt247 toxicity to nematodes (S4E Fig) [31]. We purified Cry6Ba from Bt247 cultures, but, intriguingly, the Cry6Ba toxin alone was unable to kill the host, possibly indicating that other yet unknown virulence factors are involved in Bt247 toxicity towards nematodes (S4F Fig). Indeed, Cry6Ba was previously shown to have coleopteri-cidal activity [59], but only limited nematocidal activity compared to Cry6Aa [60]. Also, Cry6 toxins are hemolysin E (HlyE)-like toxins (also known as Cytolysins A) and there are numerous examples of synergistic activities between cytolysins and other virulence factors (e.g., [61]; reviewed in [62]). In contrast to purified Cry6Ba, *C. elegans* was killed by the purified Bt679 tox-ins mixture, which includes the two nematocidal toxins Cry21Aa and Cry14Aa (S4A Fig). *elt-2* (RNAi) worms were more sensitive to the Bt679 toxin mixture than control worms (S4G Fig), pointing to *elt-2* being required for defense against Bt679 Cry toxins. To assess the separate effects of the two nematocidal Cry21Aa and Cry14Aa toxins, we took advantage of a non-patho-genic Bt679 Cry- strain that lacked these toxins [32] and re-introduced them individually. Strik-ingly, *elt-2*(RNAi) worms were susceptible to Bt679 spores, which produce soluble Cry21Aa after germination, but not to those producing Cry14Aa (S4H Fig), suggesting that *elt-2* is only required for the *C. elegans* defense response to Cry21Aa. Our results thus extend the previous findings that regulators and mediators of the *C. elegans* response vary between Cry toxins [27,29,30,39,63]. We conclude that the level of specificity in the interaction between the host and pathogenic Bt is to some extent driven by individual Bt virulence factors.

## *elt-2* RNAi leads to increased survival rate after Bt247 infection through multiple possible mechanisms

Our results on the role of *elt-2* in defense against Bt679 is in line with previously published work on other pathogens, *i.e. elt-2* regulates the expression of immune effector genes and is thus required for resistance to pathogen infection [40,42,47]. In contrast, *elt-2*(RNAi)-medi-ated increase in survival rate after infection, as we observed for Bt247, has not been reported before. Also, the high survival of Bt247 infected worms is unique among several Bt strains we tested (S4D Fig). There are several possible explanations for the observed high survival of *elt-2* (RNAi) worms on Bt247. First, ELT-2 might positively regulate the expression of genes that are required for infection and thus susceptibility to Bt247, encoding e.g. the toxin-binding receptor. Several individual host factors that are required for susceptibility to toxins Cry5B, Cry14A and Cry6A are known [39,63]. We thus tested if these host factors are involved in *elt-2* (RNAi) mediated high survival on Bt247. We could exclude an involvement of the Bt-toxin resistant (*bre*) genes required for synthesis of the Cry5B glycosphingolipid receptor [63], given that *bre* and other glycosylation-deficient mutants were susceptible to Bt679 and Bt247 infec-tion (Fig 6A). We could also exclude involvement of the necrosis pathway (Fig 6B), for which the central aspartic protease ASP-1 was previously shown to physically interact and confer sus-ceptibility to Cry6Aa [39]. The necrosis-deficient mutant *asp-1(tm666)* showed reduced sur-vival on Bt247 in one out of five runs only (Fig 6B). Also, for mutants deficient in additional

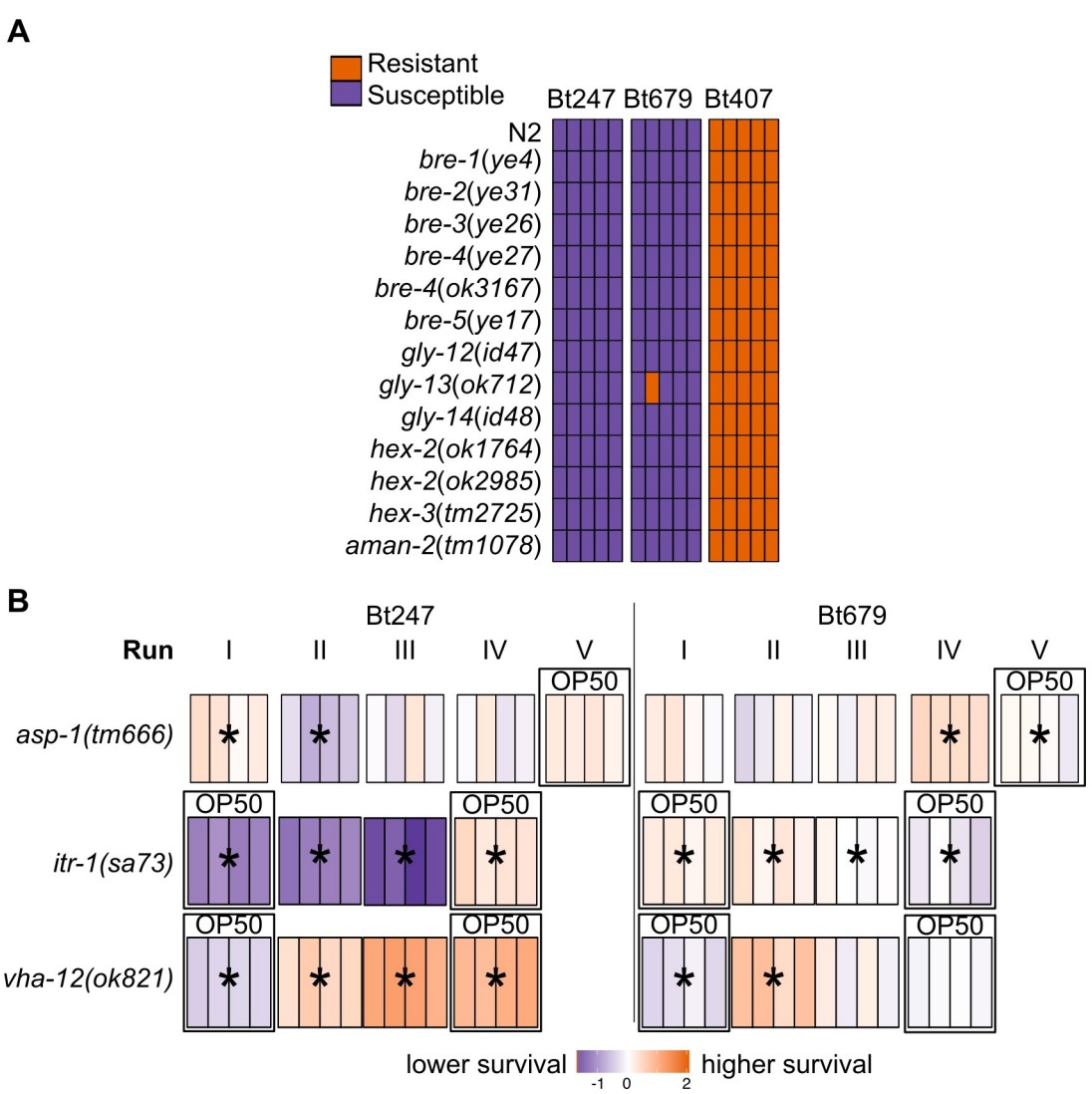

**Fig 6. Specific responses to Bt247 and Bt679 do not involve the *bre* genes or the necrosis pathway.** (A) A binary score considered resistance of glycosylation-deficient mutants to pathogen exposure as >70% survival of ~300 worms on plate. N = 5. Each bar represents the score of a single plate. Our results show that glycosylation-deficient mutants are susceptible to Bt679 and Bt247. (B) Difference in survival between the N2 control and the knockout mutants of the: terminal necrosis protein aspartase *asp-1(tm666)*, inositol triphosphate receptor ion channel *itr-1(sa73)*, and vacuolar proton translocating ATPase *vha-12(ok821)*. Asterisks show significant differences between mutant and the wildtype N2 control. Infection of Bt247 led to inconsistent survival phenotypes of the *asp-1(tm666)* mutant animals, including significantly higher survival, significantly lower survival, and no survival difference in separate runs of the experiment. Similar levels of variation were also found upon Bt247 infection of *asp-1(tm666)* mutant animals, in which *elt-2* was inactivated by RNAi. The necrosis-deficient mutant of another pathway component, *itr-1(sa73)*, had significantly higher survival rate on Bt679 and lower survival rate on Bt247 compared to control worms. The necrosis-deficient mutant of yet an additional pathway component, *vha-12(ok821)*, had higher survival rate on Bt247 compared to control worms and had inconsistent survival phenotypes on Bt679. Since knockout mutants of different necrosis pathway components show different survival phenotypes upon infection, we conclude that deficiency in necrosis is unlikely to have a strong effect on tolerance to Bt247. The data represented in the heatmap and statistics as in Fig 3E. In (B) worms were grown on RNAi *E. coli* HT115 plates or on NGM plates seeded with *E. coli* OP50 (indicated by 'OP50' above a run).

members of the necrosis pathway *vha-12(ok821)* and *itr-1(sa73)*, we were not able to find consistent results that would suggest that the necrosis pathway is involved in the response to Bt247 (Fig 6B). In this context, it is important to emphasize that our assay usually produces highly consistent results across independent biological replicates (see for example Figs 3 and 5). If in

other cases, the results are variable and not consistently negative, then this indicates that the respective gene can have at least a minor effect on the considered phenotype–in contrast to clear evidence for an absence of an effect.

We further tested the involvement of sterility, considering that sterile mutants are often resistant to pathogen infection [64–66] and that *elt-2*(RNAi) reduces worm fecundity and fertility [43]. We found that the sterile *rrf-3*(*b26*), *glp-1*(*ar202*), and *glp-1*(*bn18*) (S5A and S5B Fig) mutants had higher survival than wildtype after infection with both Bt strains and that the additional knockdown of *elt-2* further increased survival on Bt247 and displayed higher survival rate than *elt-2*(RNAi) worms alone on Bt679 (S5B Fig). We thus conclude that the increased survival rate of *elt-2*(RNAi) worms infected with Bt247 does not involve the *bre* genes and the necrosis pathway, while sterility could influence but not fully explain the increased survival rate on Bt247 and shows no strain-specific effects on survival rate. It is still possible that the increased survival rate of *elt-2*(RNAi) worms infected with Bt247 is due to a down-regulation of the unknown intestinal receptor of Cry6Ba by the *elt-2*(RNAi) treatment, since *elt-2* is a global regulator of intestinal gene expression [48]. However, unlike *bre* mutants that are completely resistant to Cry toxin exposure and do not show any intestinal tissue damage [63,67], *elt-2*(RNAi) worms infected with Bt247 show intestinal damage, despite highly efficient depletion of ELT-2 by the RNAi treatment (Figs 3F and 4M) (S2I, S2J and S2S Fig). These results suggest that in *elt-2*(RNAi) worms the Bt247 toxin conserves, at least to some extent, the capacity to cause damage and therefore that the toxin receptor is, at least to some extent, still expressed. This possibility can only be tested in more detail once the receptor of the Bt247 toxin will be identified.

Based on the previous set of results, we hypothesized that the effect of *elt-2*(RNAi) is multifactorial and ultimately leads to enhanced tolerance to Bt247 infection, as discussed above. In *C. elegans*, most studies on pathogen defense mechanisms do not differentiate between resistance and tolerance, with a few exceptions [68–70]. As a result, little is known about the underlying genetics of tolerance to infection in *C. elegans*. We thus initiated further experiments to elucidate genes possibly involved in tolerance to Bt247.

## *elt-2* regulates putative detoxification and lipid metabolism enzymes on Bt247

To further explore the molecular processes involved in the *elt-2*-mediated increase in survival rate after Bt247 infection, we performed an additional transcriptome analysis with Bt-infected *elt-2*(RNAi) worms by mRNA sequencing (S4 Table). In total, the expression of 6180 genes was affected by *elt-2*(RNAi) (Fig 7A and 7B). The strain-specific DE patterns on Bt247 and Bt679 (clusters 2 and 8 uncovered in the first transcriptome data set, Fig 2B) disappeared upon *elt-2* knockdown and the expression changes on Bt247 resembled more patterns on Bt679 (red boxes in Fig 7A). Together with our genetic analyses, these data strongly suggest that the strain-specific gene expression changes are mainly dependent on ELT-2.

We next focused on the genes commonly regulated by *elt-2* on all three Bt treatments (Fig 7B). We found 507 genes (8% DE genes) that were down-regulated in *elt-2*(RNAi) animals on all bacteria, representing the core set of genes that require *elt-2* for transcriptional activation (Fig 7B, cluster 6). Surprisingly, 698 DE genes (11% DE genes) were up-regulated in *elt-2*(RNAi) animals on all bacteria. For these genes, other TFs may overcompensate for loss of ELT-2, similar to a previously described class of genes up-regulated in the *elt-2* mutant at the first larval stage [46] (Fig 7B, cluster 9).

The genes that specifically respond to *elt-2*(RNAi) upon Bt247 infection (red box in Fig 7B, S5 Table) included 1106 down-regulated and 1578 up-regulated genes (S4 Table). The down-

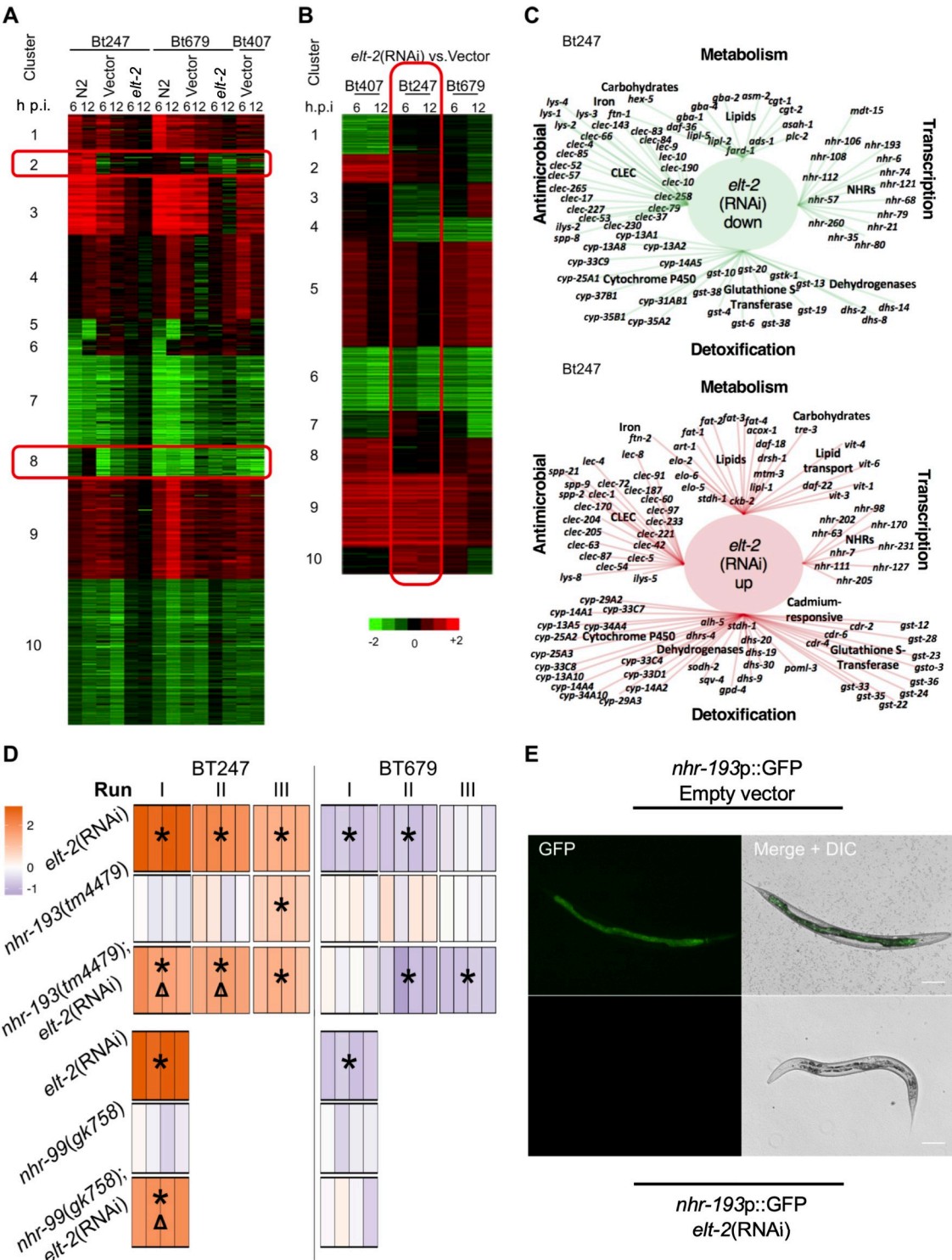

**Fig 7. Detailed transcriptional and functional analysis of *elt-2* mediated tolerance to Bt247.** (A) DE heatmap of a combined analysis of the first transcriptome dataset in wildtype N2 animals (first two columns of each pathogen strain section; subset of data from Fig 2) and a second additional transcriptome data set using *elt-2*(RNAi) worms and empty vector controls 6 h and 12 h p.i. with Bt247, Bt679 or the non-pathogenic Bt407. Clusters including genes commonly up- and down-regulated on all bacteria are highlighted. (B) Heatmap of the same data, highlighting DE between *elt-2*(RNAi) and empty vector controls on each Bt strain. (C) Gene ontology (GO) functional enrichment analysis of genes up- and down-regulated (as indicated) when comparing *elt-2*(RNAi) worms with controls, both exposed to Bt247. For both gene sets, the GO terms metabolism, transcription, detoxification and

antimicrobial were among the most significantly enriched functions. Examples of genes supporting these enriched terms are shown. (D) Survival of the nuclear hormone receptor mutants *nhr-193*(*tm4479*) and *nhr-99*(*gk758*), exposed to Bt247 and Bt679 and epistasis analysis following *elt-2*(RNAi). Heatmaps and statistics as in Fig 3E. Triangles indicate a significant difference compared to *elt-2* (RNAi). (E) Expression pattern of the transcriptional reporter strain *nhr-193*p::GFP (MY1094) in empty vector control worms and *elt-2*(RNAi) L4 worms. Scale bar represents 50 μm. See also S6 Fig.

regulated gene set was significantly enriched for genes putatively involved in lipid metabolism, detoxification, xenobiotic metabolism, and also genes encoding C-type lectins and collagens (Fig 7C, S5 Table). Since their down-regulation is associated with increased survival rate after Bt247 infection and possibly with tolerance to infection, activity of these processes and genes may be required by some part of the Bt infection process and thus their inactivation prevents Bt-mediated killing. The down-regulated category of lipid metabolism includes genes predicted to function in sphingolipid synthesis and breakdown [71], such as the ceramide glucosyl transferase genes *cgt-1* and *cgt-2*, the glycosylceramidases *gba-1*, *gba-2*, and *gba-4*, the sphingomyelin phosphodiesterase *asm-2*, and the acylsphingosine amidohydrolase *asah-1*. Notably, CGT-1, CGT-2, and CGT-3 were previously shown to add carbohydrate groups to the sphingolipid ceramide [72], yielding glucosylceramide, which can be further glucosylated by BRE-2, BRE-3, BRE-4, and BRE-5, leading to glycosylated sphingolipids, which are known targets of the Bt toxins Cry5B and Cry14A [63]. As *bre* mutants are susceptible to Bt247 (Fig 6A), we speculate that glucosylceramides also act as precursors for other glycosphingolipids that may serve as receptors for Cry6B. Moreover, the down-regulated gene set included two transcriptional regulators of *C. elegans* fatty acid metabolism, the transcriptional mediator subunit MDT-15 and the nuclear hormone receptor NHR-80 [73], indicating a role of fatty acid metabolism in Bt247 tolerance (see below).

The set of up-regulated genes was similarly enriched for detoxification and lipid metabolism, although based on different genes than those in the down-regulated set (Fig 7C, S5 Table). The "lipid metabolism" category contained several genes involved in fatty acid desaturation and elongation, including the desaturase enzyme genes *fat-1*, *fat-2*, *fat-3*, and *fat-4*, and the fatty acyl elongase *elo-2*. Since their up-regulation correlated with increased survival, we propose that alterations in fatty acid content and abundance of poly-unsaturated fatty acids (PUFAs) might restrict the effects of Bt247 infection. PUFAs play important roles in membrane structure and function and are precursors for a variety of different signaling molecules and ligands for transcription factors [74]. Additional work will be required to evaluate whether and how exactly fatty acid composition and PUFAs can influence a defense trait like tolerance.

### *elt-2* RNAi-mediated high survival on Bt247 is influenced by the nuclear hormone receptors NHR-193 and NHR-99

To explore the possible role of fatty acid metabolism in increased survival rate after infection with Bt247 and possibly tolerance, we focused on potential regulators such as nuclear hormone receptors (NHRs) [75]. Interestingly, the Bt247-exclusive responsive genes in cluster 2 from our first transcriptome data set (Fig 2B) were significantly enriched for *nhr-8* targets (S3 Table). Moreover, *nhr-8*, *nhr-99*, and *nhr-193* were all part of cluster 2 and *nhr-193* expression was down-regulated upon *elt-2*(RNAi) (Fig 7D, S4 Table). NHR-8 regulates genes required for elongation and desaturation of fatty acids and also the expression of xenobiotic detoxification genes [76,77], while the function of NHR-99 and NHR-193 is unknown. We assessed the potential link between ELT-2 and these NHRs in the context of Bt infection and first exposed knockout mutants of these genes to Bt247 and Bt679. While there was no effect for *nhr-99* (*gk758*) and *nhr-193*(*tm4479*), *nhr-8*(*ok186*) worms survived significantly better than wildtype

N2 on both Bt strains (Fig 7D and S6 Fig). Thus, NHR-8 contributes to the general but not to the strain-specific Bt response. *nhr-8*(*ok186*);*elt-2*(RNAi) worms did not vary in survival to the *elt-2*(RNAi) control (S6 Fig), while both *nhr-99*(*gk758*);*elt-2*(RNAi) and *nhr-193*(*tm4479*);*elt-2* (RNAi) worms had lower survival rate than *elt-2*(RNAi) worms on Bt247 (Fig 7D). Importantly, the knockout of *nhr-99* and *nhr-193* had no significant effect on the survival rate of elt-2(RNAi) worms after Bt679 infection, supporting their specific role towards Bt247 only (Fig 7D). Using a transcriptional GFP reporter for *nhr-193*, we found strong, *elt-2*-dependent expression in the cytoplasm and especially nuclei of intestinal cells (Fig 7E), consistent with a function in the worm's gut. Altogether, our results suggest that *elt-2*(RNAi)-mediated increase in survival rate after Bt247 infection is partially dependent on NHR-99 and NHR-193.

### A model of *elt-2* mediated distinct immune responses to Bt247 and Bt679

We used the nematode *C. elegans* as a model to characterize common and distinct responses against different strains of the same pathogen species. We were interested in understanding in how far differences in virulence factor expression and infection dynamics, caused by closely related pathogens, could elicit distinct defense responses in an invertebrate host. We combined several transcriptome datasets with functional genetic analysis of candidate genes and identified the GATA TF ELT-2 as central regulator of pathogen strain-specific responses. Surprisingly, silencing of *elt-2* caused contrasting effects towards the two tested pathogenic Bt strains. *elt-2* was previously proposed as the central general regulator of defense responses against intestinal pathogens [17,41], implying that it mediates similar responses to a large diversity of pathogen taxa. Yet, it was unknown that it could additionally be involved in highly fine-tuned responses to two strains of the same pathogen species. In particular, *elt-2* is required for *C. elegans* resistance to Bt679, most likely through activating specific effector proteins such as SPP-8 and SOD-3 and acting in parallel to the p38 MAPK pathway (Fig 8). In contrast, *elt-2* inactivation causes increased survival rate after infection with Bt247 (Fig 8), which is characterized by reduced intestinal damage despite a high pathogen burden. *elt-2*(RNAi) thus seems to have an impact on tolerance, rather than resistance to Bt247 infection. The *elt-2*(RNAi)-mediated response to Bt247 infection is completely independent of the p38 MAPK pathway, which to date has been associated with resistance to multiple pathogen threats [78,79]. Moreover, *elt-2* (RNAi)-mediated increase in survival rate after Bt247 infection seems to be multifactorial. Gene expression analysis points to a possible involvement of multiple candidate mechanisms such as detoxification and lipid metabolism, which need to be validated in future studies. Further, we show that *elt-2*(RNAi) mediated survival on Bt247 partially depends on two additional TFs, encoded by the nuclear hormone receptor genes *nhr-99* and *nhr-193*. Overall, our study provides new insight into the molecular mechanisms underlying distinct defense responses to infection with two different pathogen strains of the same species in an invertebrate. As these pathogen strains produce distinct virulence factors that however have a similar mode of causing damage in that they form pores in the membrane, our study further highlights an unexpected complexity of invertebrate defense strategies against pathogens. We anticipate that our findings may stimulate similar studies on pathogen-strain-specific host responses, which are so far largely unexplored in other invertebrate taxa.

## Materials and methods

### *C. elegans* and bacterial strains

Worm strains were maintained on nematode growth medium (NGM) plates at 20°C and fed with *E. coli* OP50 as described in [80]. Strains used and the sources from which they were obtained are listed in S6 Table. Bt spore solutions were obtained as previously described

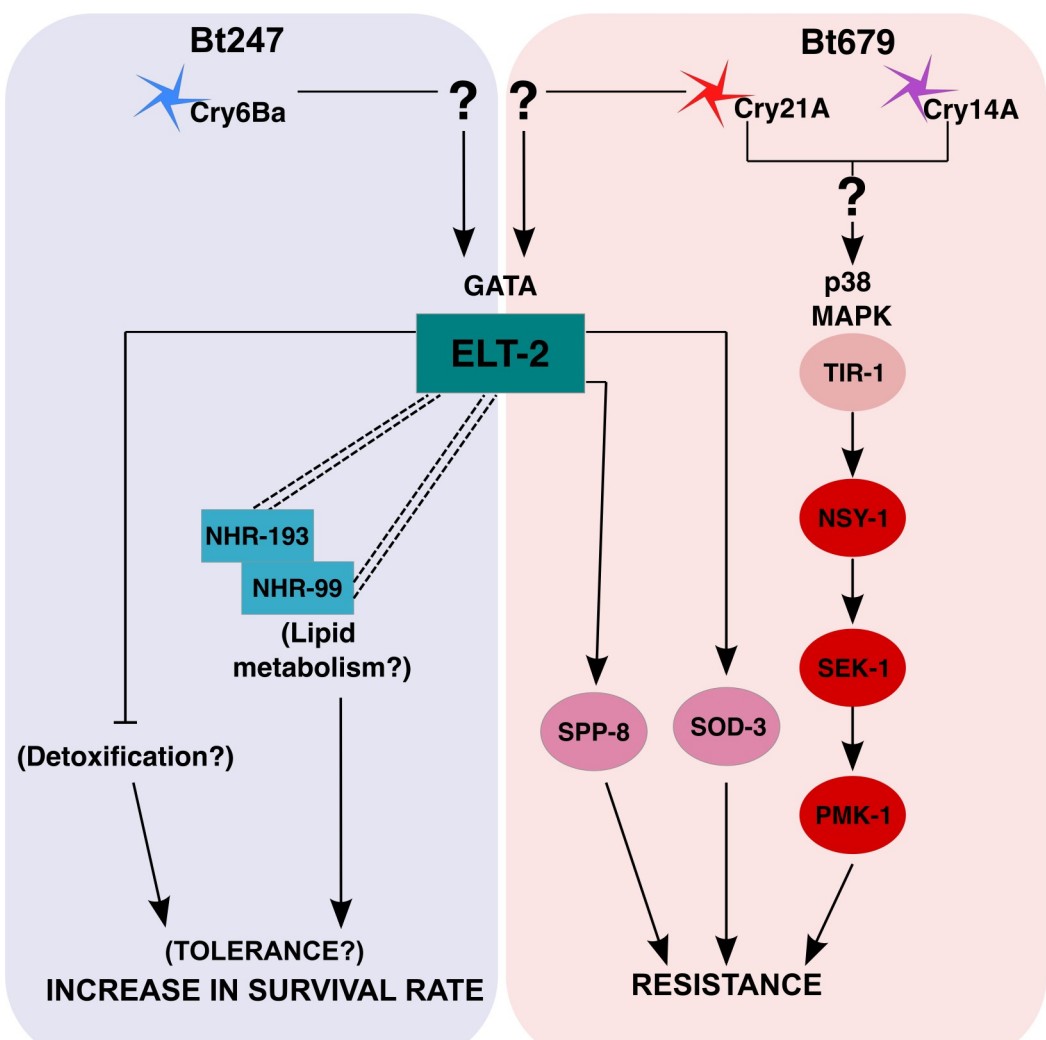

**Fig 8. Model of the host genes and processes involved in Bt strain-specific interaction with *C. elegans*.** ELT-2 plays a central role in mediating distinct responses to Bt247 (blue background) and Bt679 (red background), as shown by the differential regulation of known ELT-2 targets by Bt247 and Bt679 exposure and the disappearance of this strain-specific DE pattern upon *elt-2* knockdown in our transcriptome analyses. Other transcription factors such as NHRs appear to interact with *elt-2* to hone the transcriptional response to a specific pathogen. Please note that while only strain-specific response pathways are shown here, the responses to the two Bt strains are not completely distinct, common response pathways like the one mediated by *jun-1* may interact or act in parallel to ELT-2. The differential activation of the ELT-2 targets must be initiated by an upstream sensor, which is represented here by question marks and still needs to be identified in future work. Arrows represent positive influence in the function of a given gene or process, bars represent a negative influence and double dashed lines represent genetic interaction between genes or biological processes.

[22,81]. Briefly, we thawed frozen stocks on Luria-Bertani (LB) plates, which were incubated overnight at 25˚C. Sterile disposable inoculation loops (Roth) were used to pick a single colony of Bt407, or multiple colonies of Bt679 and Bt247 (to ensure that a colony carrying the Cry toxin-encoding plasmids was selected), to inoculate 1000 mL of sterile liquid Bt medium (Bacto-peptone (Sigma)(7.5 g/L), glucose (5.56 mM), $KH_2PO_4$ (22.06 mM), $H_2HPO_4$ (22.99 mM), pH adjusted 7.2) supplemented with 5 mL filter-sterilized salt solution ($MgSO_4.7H_2O$ (100 mM), $MnSO_4.H_2O$ (2.37 mM), $ZnSO_4.7H_2O$ (9.76 mM), $FeSO_4.7H_2O$ (14.39 mM)) and 1250 μL $CaCl_2$ (1M). Cultures were incubated at 28˚C with shaking for seven days. After four days of incubation the cultures were supplemented with additional 5 mL salt solution (as

previously described) and 1250 μL CaCl$_2$ (1M). The described culturing method is used to induce sporulation. Cultures containing a spore-toxin-particle mix were subsequently harvested by centrifugation in 50 mL Falcon tubes (SARSTEDT) for 10 min at 4000 rpm. Supernatants were removed, pellets were resuspended in Phosphate Buffered Saline (PBS) and particle concentrations were assessed with the Neubauer improved counting chamber. Culture concentrations ranged between 109–2 x 10$^{10}$ particles/mL for strains Bt247, Bt247 Cry-, Bt679, Bt679 Cry- (RFP), Bt679 Cry- (Cry14Aa) and Bt679 Cry- (Cry21Aa) and between 10$^3$–10$^4$ particles/mL for Bt407. The resulting cultures were aliquoted in 1.5 mL Eppendorf tubes and stored at -20˚C for further use.

## Bt survival assays

Bt survival assays were performed as previously described [18]. Briefly, to assess survival of wildtype (N2) and mutant strains, worm populations with many eggs were washed off plates with M9 buffer (KH$_2$PO$_4$ (22 mM), Na$_2$HPO$_4$ x 2 H$_2$O (33.7 mM), NaCl (85.6 mM), supplemented with 1 mL MgSO$_4$ (1M) after autoclaving) and synchronized by bleaching: 4 mL washed worms/eggs in M9 were mixed in a 15 mL Falcon tube (SARSTEDT) with 1 mL of a 1:1 solution of NaOH (5M) and NaClO (12%) for 5 min with vortexing and inverting. Bleaching was stopped by centrifuging for 1 min at 3500 rpm, supernatant was discarded, and pellet was washed three times with 5 mL M9 buffer. Only eggs survived the treatment. An extra synchronization step included incubation of bleached eggs in M9 buffer overnight in a horizontal shaker at 20˚C until all eggs had hatched. Hatched larvae were raised on 9 cm diameter NGM plates seeded with 700 μL of an overnight liquid culture of *E. coli* OP50 in LB broth medium. At the L4 larval stage approximately 30 worms per plate were transferred by pipetting to 6 cm diameter peptone-free NGM (PFM) plates inoculated with either 75 or 100 μl of Bt spore solutions mixed with *E. coli* OP50 adjusted to an optical density (OD$_{600}$) of 5 with PBS buffer. Different dilutions of pathogenic Bt spores were tested to produce dose-response curves. Non-nematocidal Bt strain Bt407 Cry- (Bt407) lacks pore-forming toxins and was used as infection treatment control, but only at the single highest concentration in which pathogenic Bt were tested. The infection plates were left to dry overnight at 20˚C. Alive and dead worms were assessed by reaction to gentle touch with a platinum wire at 24 h p.i. Analysis of survival experiments was done using R [82] and final preparation for visualization was done using Inkscape (http://www.inkscape.org/). The model fitted was glm(formula = cbind(Worms Alive, Worms Dead) ~ worm treatment (RNAi) and/or worm strain*Bt concentration, family = binomial). Statistical significance of treatment effects was assessed post hoc using Tukey HSD test and *p*-values were Bonferroni corrected.

## RNA interference

For *elt-2* knockdown, worms were fed with the X-5I11 *E. coli* HT115 RNAi clone from the Source BioScience Ahringer RNAi library [83]. *E. coli* HT115 containing L4440 RNAi empty vector were used as negative control. As positive control of gene knockdown a subset of worms was fed in parallel with Ahringer *E. coli* HT115 RNAi clones I-1A05 *bli-3* (multiple blisters), IV-6K06 *unc-22* (twitching), and II-7O15 *plc-3* (severely reduced egg laying). 300 synchronized L1 larvae were placed on 6 cm plates seeded with 100 μl of concentrated RNAi bacteria for 48 hours. Survival analysis was done as described above. ELT-2 is a major regulator of intestinal development. To verify that the changes in the survival phenotype of *elt-2*(RNAi) worms exposed to pathogenic Bt were not exclusively due to an abnormal development we also performed *elt-2* knockdown starting at the L4 larval stage for 48 hours. Three-day control and *elt-2*(RNAi) adult worms were exposed to Bt and survival was scored as above.

## Bt exposure, RNA isolation and differential expression analysis

Two transcriptomic studies were carried out. For the first transcriptomic study N2 worms at the L4 larval stage were exposed to Bt as described before in 9 cm diameter PFM plates inoculated with 250 μl of Bt spore solution diluted with *E. coli* OP50 OD$_{600}$ 5. Worms were exposed to an equal dose of spores of the pathogenic strains Bt247 (leading to 88% survival 24 h p.i.) and Bt679 (leading to 50% survival 24 h p.i.) and of the non-pathogenic controls Bt407 and *E. coli* OP50. After 2 h, 6 h, 12 h and 24 h p.i worms were washed off the plates using PBS supplemented with Tween-20 to a final concentration of 0,3% (PBST). Worms were centrifuged and washed with PBST buffer 3 times and then transferred to TRIzol lysis Reagent™ (Invitrogen). Worms were cyclically thawed and frozen five times using liquid nitrogen and a heating block at 37°C to allow the TRIzol reagent to enter the cells and better preserve the RNA. RNA was extracted using the NucleoSpin RNA purification kit (Macherey-Nagel), treated with DNAse and stored at -80°C. Five completely independent replicates were prepared in parallel and three were chosen for further sequencing. Messenger RNA samples were indexed and sequenced using the Illumina HiSeq 2000 technology to obtain paired-end 100 bp-long reads with a coverage of 20 million reads per sample.

The bioinformatics analyses were as followed: reads were aligned to the *C. elegans* genome form Wormbase version WS235 (www.wormbase.org) by Tophat2 [84] (ccb.jhu.edu/software/tophat/index.shtml), followed by differential expression analysis using Cuffdiff [84] (cole-trapnell-lab.github.io/cufflinks/cuffdiff/) and ABSSeq [85]. K-means clustering analysis was done using Cluster 3.0 [86] (bonsai.hgc.jp/~mdehoon/software/cluster/software.htm) with a k of 10 for both experiments. A heatmap was created using TreeView version 1.1.4r3 [87]. Promoter region motif enrichment analysis was done on the promoter regions -600 bp and 250 bp relative to transcription start sites of the genes in each cluster with the software AMD [88]. Enrichment analysis of each cluster or groups of up- and downregulated genes was done using EASE [89], DAVID [90,91] and WormExp [92] (wormexp.zoologie.uni-kiel.de/wormexp/).

For the second transcriptomic study all procedures and analyses were as in the first transcriptomic study except that control and *elt-2* knockdown worms treated with RNAi bacteria for 48 h starting at the L1 larval stage were exposed to different concentrations of pathogenic Bt247 and Bt679. Since the first transcriptomic study established that there were no significant differences in expression between *E. coli* OP50 and the non-pathogenic Bt407, a single concentration of Bt407 equivalent to the highest used for pathogenic Bt strains was used. Samples were collected 6 h and 12 h p.i, since these were the time points with the most significant strain-specific expression changes in the first transcriptomic study. Survival rate after 24 h p.i. was assessed for all treatments and those with 70% survival on pathogenic Bt strains were chosen for sequencing. As in the first transcriptomic study five completely independent replicates were prepared in parallel and three were chosen for further sequencing. Bioinformatic analyses were as in the first transcriptomic study.

## Bacterial load assessment by counting colony forming units (CFUs)

Bacterial load was estimated by exposing the worms as described for the survival assays and at 12 h p.i. five worms were picked to 50 μL M9 buffer supplemented with 25 mM tetramisole (TM buffer) and washed two times with TM buffer supplemented with 100 μL/mL gentamicin and two additional times with M9 buffer. After washing, worms were resuspended in 150 μL PBS supplemented with 1% Triton X-100, zirconia beads (1 mm in diameter) were added, and worms were ground using a SPEX SamplePrep 2000 GenoGrinder Tissue Homogenizer at 1200 strokes/min for 3 minutes. The worm lysate was serially diluted in PBS and the dilutions 1:100 and 1:1000 were plated on LB agar in triplicate and incubated at 20°C for 48 h for colony

counting. The buffer from the last wash was also plated in triplicate to assess washing efficiency. Five worms from each treatment were mounted onto microscope slides and their length was measured using a Leica M165 FC Fluorescence Stereomicroscope and the LAS software. Worm length was used to normalize the bacterial load per mm of worm as *elt-2*(RNAi) and empty vector worms have different lengths.

## Fluorescence, light microscopy and transmission electron microscopy (TEM)

For fluorescence microscopy *elt-2*(RNAi) worms were exposed to Bt in 6 cm diameter plates and at each time point 10 worms were transferred to a microscopic slide with an agar pad, immobilized with 50 mM sodium azide and observed with a Carl Zeiss Observer Z.1 inverted fluorescence microscope or the confocal Carl Zeiss LSM 700. For light microscopy, samples were observed under the Carl Zeiss Observer Z.1 microscope. For TEM, worms were exposed to Bt as described for RNA extractions and washed off plates 12 h p.i. directly with fixation solution (2.5% glutaraldehyde, 2% paraformaldehyde, 0,1 M cacodylate buffer pH 7.2). Worms were cut in half to ensure penetration of the fixation solution and prepared as described in [93] for visualization with a Tecnai 2 electron microscope in the microscopy facility of the Biology Tower at the Christian-Albrechts University Kiel.

## Western blot

1000 worms were prepared on 9 cm NGM plates and exposed to bacteria as described for survival experiments. 6 h p.i. worms were washed with M9 buffer and protein extraction was done using standard methods [94]. Briefly, around 150 to 3000 worms were collected in 2 mL safe-lock Eppendorf tubes and centrifuged at 500 rpm for 1 min. Supernatant was discarded and samples were resuspended in lysis buffer (for 500 μL lysis buffer use: 5 μL proteinase supplemented with phosphatase inhibitor (ThermoFisher), 5 μL glycerol, 50 μL sodium deoxycholate (10%), 1.1 μL ethylenediaminetetraacetic acid (EDTA) (0.45M), 438,9 μL buffer (for 500 mL buffer use: 3.0 g Tris pH 8, 4.4 g NaCl, 5 mL Triton X-100, complete to volume with deionized water)). Samples were cycled seven times between freezing in liquid nitrogen and thawing at 37°C with vortexing in between. After cycling, samples were incubated on ice for 10 min and then centrifuged at 13,000 rpm for 10 min at 4°C. Supernatant was pipetted to a new 1.5 mL Eppendorf tube and stored at -20°C for later use. Protein samples were probed with anti-ELT-2 monoclonal antibody provided by James McGhee, University of Calgary, AB, Canada, and anti-actin clone C4 monoclonal antibody (Millipore, Cat. # MAB1501R, Lot # 2387816), followed by goat anti-mouse-IgG (H+L) HRP conjugate (Advansta, Part number: R-05071-500, Lot number: 110922–26). PageRuler Plus prestained ladder (Thermo Fischer, Product # 26619, Lot # 00119113) was used. Samples were visualized using SuperSignal West Pico Chemiluminescent substrate (Thermo Scientific, product #34080).

## Intestinal integrity assay (Smurf)

The smurf worm protocol was followed as previously described [95]. Briefly, worms were infected as described in the Bt survival assay section and after 24 h p.i. they were washed with 5 mL S buffer (NaCl (100 mM), $K_2HPO_4$ (6.5 mM), and $KH_2PO_4$ (43.5 mM)), spun down and resuspended in an overnight liquid *E. coli* OP50 culture mixed with blue food dye (Erioglaucine disodium salt, Sigma-Aldrich; 0.05g/mL) for three hours. Animals were then washed three times with S buffer, centrifuged at 3500 rpm for 1 min, and 20 μL of the pellet was mounted on microscopic slides with 2% Agarose pads. 10 μL Sodium azide (20 mM) was added to paralyze the worms for imaging and scoring 30 worms for leaking and non-leaking

phenotypes as shown in S2Q and S2R Fig. Samples were visualized using a Zeiss Observer Z.1 microscope and a Leica M165 FC Fluorescence Stereomicroscope.

## Crystal production & purification

The three different Bt strains Bt679, Bt247 and Bt247(-Cry6Ba) were grown on LB-agar plates. For each strain, one colony was picked to inoculate an overnight 5 mL LB liquid preculture. Precultures were spread on T3 sporulation medium (per liter: 3g tryptone, 2 g tryptose, 1.5 g yeast extract, 0.05 M phosphate buffer pH 6.8, and 0.005 g of $MnCl2$) and incubated at 30˚C for 4 days to promote Bt sporulation and toxin crystal production. Spores and crystals were collected using a cell scraper, resuspended in water and centrifuged once at 9500 rpm for 45 min. The pellet was resuspended in water. For each strain, an aliquot of unpurified crystals was kept at 4˚C for controlling the presence of toxins and as a quality control of the purification step. Crystals from Bt679 and Bt247 strains were purified using a discontinuous sucrose gradient (67-72-79%) using a SW-32 Ti swinging-bucket rotor on an Optima XPN ultracentrifuge (Beckman Coulter). After 16 h of ultracentrifugation at 23,000 g and 4˚C, the crystals were recovered from the 67–72% interphase for Bt247 and from both the 67–72% and 72–79% interphases for Bt679. Spores, which formed a tight pellet at the bottom of the tubes, were discarded. Crystals were then washed by performing several rounds of centrifugation and resuspension in water to discard as much sucrose as possible. Purified crystals were stored in water at 4˚C until use.

## Quality control of crystal productions

Quality control of crystal products was done as described in [96]. Briefly, unpurified productions and purified crystals suspensions were verified on SDS-PAGE 12% gels. Laemmli buffer containing 10 mM dithiothreitol (DTT) was added to the samples, which were then heated at 95˚C for 5 min prior to loading on the gel. Migration was performed at 110V for 2 h and gels were stained overnight using Instant*Blue* Coomassie straining (Sigma Aldrich). The gels were subsequently washed several times with water and digitalized using a ChemiDoc XRS+ and ImageLab software version 6.0.0 (Bio-Rad). Intensity of the peaks corresponding to the different Bt679 toxins were analyzed with ImageJ software v1.51k [97]. Purified Bt247 crystals were additionally analyzed by MALDI-ToF mass spectrometry using an Autoflex mass spectrometer (Bruker Daltonics, Bremen, Germany) operated in linear positive ion mode. External mass calibration of the instrument, for the 10–70 kDa m/z range, was carried out using protein calibration standard II from Bruker Daltonics. Bt247 crystals were dissolved in an acetonitrile (ACN)-water mixture (70:30, v/v) and then mixed at different ratios with a sinapinic acid (SA) matrix (Sigma; 20 mg mL-1 in water/ACN/trifluoroacetic acid, 70:30:0.1, v:v:v) to obtain the best signal-to-noise spectra. Considering that all ratios gave similar results, only the 1:5 (v:v) ratio is shown. 1–2 μL of the mixture were deposited on the target and allowed to air dry. Mass spectra were acquired in the 10 to 160 kDa m/z range and data processed with Flexanalysis software (v.3.0, Bruker Daltonics).

## Generation of transgenic *C. elegans* strains

The transcriptional reporter *nhr-193*p::GFP contains 658 bp of genomic region upstream of the *nhr-193* start codon. It was generated by fusion PCR [98] using the primers SHP860, SHP861 and SHP862 (S7 Table), and injected into *unc-119*(*ed3*) animals at a concentration of 10 ng/μl together with the co-injection marker *myo-2*p::RFP at a concentration of 25 ng/μl, and the *unc-119* rescue construct pPK605 (Addgene) at 20 ng/μl, yielding one transgenic line, MY1094 [*unc-119*(*ed3*); *yaEx80*[*pnhr-193*::GFP; *myo-2*p::RFP; pPK605]], which was analyzed

here. pPK605 was a gift from Patricia Kuwabara (Addgene plasmid # 38148; http://n2t.net/addgene:38148; RRID:Addgene_38148).

All phenotypic raw data used to produce the figures in the main text and supplemental material can be found in S8 Table. The results of the statistical tests performed on the phenotypic raw data, as indicated in the figure legends, can be found in S9 Table.

## Supporting information

**S1 Fig. The opposite *elt-2*(RNAi) survival phenotype is not due to differences in Bt strain virulence and is not affected by the intestinal GATA TF *elt-7*. Related to Fig 3.** (A) Survival of *elt-2*(RNAi) (dashed lines) and RNAi control worms (solid lines) 24 h p.i. with Bt247 and Bt679. The figure represents the same data that are shown in Fig 3C and 3D and in run I of Fig 3E but here they are plotted in the same coordinate axes. In this way it is visible that Bt679 is able to kill the host to the same extent as Bt247 but at lower concentration, showing that Bt679 is more virulent than Bt247. (B) Survival of *elt-2*(RNAi) and RNAi control worms 24 h p.i. with different cultures of Bt247 and Bt679 compared to (A). Both Bt strains kill the same proportion of hosts at the same concentration, showing that they have the same level of virulence. The opposite survival phenotype is also observed in *elt-2*(RNAi) worms when they are exposed to Bt247 and Bt679 with the same virulence levels. Mean and SEM are shown, N = 5 plates with 30 worms each. Statistics as in Fig 3E. *** shows p-value < 0.001 comparing RNAi to controls. Bonferroni adjusted. (C) *elt-7* knockout mutants do not exhibit a Bt strain-specific survival phenotype. Heat maps show difference in survival at 24 h p.i. between wildtype N2 and *elt-7*(*tm840*) and *elt-7*(*ok835*) mutant animals. As in Fig 3, the shadowing between survival curves of *elt-2*(RNAi) and controls in (A-B) reflects higher or lower survival of treatment compared to control as indicated in the color scale bar of (C).
(TIF)

**S2 Fig. *elt-2*(RNAi) worms maintain integrity of the intestinal epithelium during Bt247 infection. Related to Fig 4. (A-P)** Fluorescence images and fluorescence merged with bright field images of the strain BJ49 carrying an IFB-2::CFP transgene 24 h after exposure to (A-D) *E. coli* OP50 (E-H), Bt407 (I-L) Bt247, and (M-P) Bt679. Scale bars represent 100 μm. IFB-2 is a structural component of the intestinal terminal web [99]. The CFP tag surrounds the intestinal lumen and is seen as two parallel thin continuous lines when healthy worms are inspected under the fluorescence microscope [99]. Damage in the intestinal epithelium can be observed when these continuous thin lines produce gaps, twists, blur or collapse into one line [100]. Consistent with the results of the TEM analysis (Fig 4C–4N), we found that low survival rate of *elt-2*(RNAi) worms after Bt679 infection coincides with exacerbated damage to the intestinal brush border and terminal web (S2O and S2P Fig) compared to empty vector controls (S2M and S2N Fig). However, higher tolerance of *elt-2*(RNAi) to Bt247 (S2I and S2J Fig) infection coincides with substantially less damage at the intestinal epithelium compared to control worms (S2K and S2L Fig). Worms exposed to non-pathogenic bacteria show no damage to their intestinal brush border and terminal web at 2 h p.i., regardless of the *elt-2*(RNAi) treatment (S2A–S2H Fig). **(Q-S)** Smurf assay to assess intestinal epithelium integrity [95]. At 24 h p.i. worms were exposed for 3 hours to *E. coli* OP50 with blue food dye. We observed if there was leaking of the dye from the intestinal lumen into the intestinal cells as a proxy for disruption of intestinal integrity. We show a representative picture of (Q) a non-leaking intestine (integral), (R) a leaking intestine (loss of integrity or damage), respectively indicated by arrows, and (S) the relative frequency of intestinal damage of worms exposed to Bt679, Bt247 and bacterial controls. Damage was scored as a binary observation (present/absent) of blue food dye leaking inside intestinal cells as opposed to remaining contained in the intestinal lumen. 30

worms per treatment were observed and the relative frequency was calculated as the number of worms presenting damage over the total number of worms per treatment. Mean and SEM are shown N = 3, Mann Whitney U test Bonferroni adjusted, *** indicates p < 0.001. We found that *elt-2*(RNAi) worms infected with Bt679 showed no difference, whereas those infected with Bt247 showed significantly lower damage than the empty vector controls (S2S Fig).
(TIF)

**S3 Fig. Subset of the survival curves used to construct the heatmaps in Fig 5A. Related to Fig 5**. (A and B) Survival of wildtype N2, *pmk-1*(*km25*) mutant, *elt-2*(RNAi), and *pmk-1* (*km25*);*elt-2*(RNAi) animals 24 h p.i. with (A) Bt679 and (B) Bt247. In A and B dashed lines represent worms with *elt-2*(RNAi) treatment. Mean and SEM are shown, N = 5 plates with 30 worms each. The figure represents the same data that are shown in Fig 5A run I. Statistics as in Fig 3E. *** shows p-value < 0.001 comparing RNAi to empty vector, +++ shows p-value < 0.001 comparing mutant to N2. Bonferroni adjusted. Green dots in (A-B) represent survival of worms exposed to Bt407. Our results suggest that the p38 MAPK pathway is required for the *C. elegans* defense to Bt679, but dispensable for the defense response to Bt247.
(TIF)

**S4 Fig. Variation in *elt-2* mediated effects upon exposure to different Bt toxins. Related to Fig 5.** SDS-PAGE gel (12% of samples loaded with Laemmli buffer and DTT, heated five minutes at 95˚C, run at 110V for 2 h, and stained with InstantBlue overnight) showing (A) purified crystal bands from a culture of Bt679, including the nematicidal Cry toxins Cry14Aa and Cry21Aa, and (B) purified Bt247 toxin, showing one unique purified crystal band between 40 and 50 kDa (expected from sequence: 44 kDa) corresponding to Cry6Ba (not present in Bt247 -Cry6Ba). (C) MALDI analysis of the Bt247 toxin, with 34 μM (1,5mg/mL) of sample re-suspended in 100 μl water (70:30) and matrix SA dd (200 mg/mL; I/H$_2$0/TFA 70:30:0,1), confirms that the major crystal produced is a monomer of 44 kDa (expected for Cry6Ba). There are also different charges for the same toxin (bicharged = 44 kDa/ 2 charges = 22). Similarly, a small amount of dimer is detected (91 kDa). (D) *elt-2*(RNAi) worms are more susceptible to infection with the BT strains Bt243, Bt245, and Bt246. Difference in survival between N2 control and *elt-2*(RNAi) animals 24 h p.i. with additional nematicidal strains Bt243, Bt245 and Bt246. (E) Survival of N2 wildtype control animals 24 h p.i. with either wildtype Bt247, expressing the Cry6Ba toxin, or Bt247(-Cry6Ba), which lacks the toxin genes and was thus unable to kill *C. elegans*. (F) Boxplots show survival of N2 wildtype control animals 48 h p.i. with purified Cry6Ba toxin resuspended in PBS or in spore solutions of the indicated Bt strains. Purified Cry6Ba diluted with PBS or mixed together with Bt spores at a range of concentrations between 1:2 to 1:1000000 was not able to kill worms. (G) difference in survival between N2 wildtype controls and *elt-2*(RNAi) animals 24 h after exposure to different dilutions of a mix of crystal toxins purified from a culture of Bt679. (H) *elt-2* is required for survival after exposure to the Bt679 PFT Cry21Aa3 but not Cry14Aa2. *elt-2*(RNAi) worms exposed to Bt679 (-Cry) expressing either Cry14Aa2 (Cry14A), Cry21Aa3 (Cry21A) or RFP. Evolved Bt247 and Bt679 strains that have lost their Cry toxin genes are unable to kill nematode hosts, confirming that Bt247 and Bt679 Cry toxins are essential virulence factors (S3E and S3H Fig (RFP, green dots)). In E, G and H mean and SEM are shown, N = 4 (except for H, where N = 3) plates with 30 worms each. Statistics as in Fig 3. *** shows p-value < 0.001 comparing RNAi to wildtype N2 or empty vector controls. Bonferroni adjusted. Green dots in (E) represent survival of worms exposed to Bt407 and in (H) survival of worms exposed to Bt679(-Cry)+RFP. In D, G and H data represented in heatmap and statistics as in Fig 3E. We conclude that the level of specificity in the nematode's response to Bt is not only determined by the pathogen strain, but

at least to some extent by individual virulence factors.
(TIF)

**S5 Fig. Sterility does not show pathogen strain-specific effects. Related to Fig 6.** In (A, B) we show the difference in survival between the N2 control and (A) the sperm-deficient and sterile RNA-dependent RNA polymerase *rrf-3*(*b26*) mutant; (B) the LIN-12/Notch family of receptors member *glp-1*(*ar202*) and *glp-1*(*bn18*) sterile mutants. Triangles indicate significant difference compared to *elt-2*(RNAi). Sterile mutants had higher survival than controls both on Bt247 and Bt679, suggesting that the effect of sterility in our infections is not Bt strain specific. Knockdown of *elt-2* in the *rrf-3*(*b26*), *glp-1*(*ar202*), and *glp-1*(*bn18*) mutant backgrounds further enhanced the increased survival rate observed in sterile mutants infected with Bt247 and produced higher survival rate on Bt679 than that of *elt-2*(RNAi) worms alone. Our results suggest that the increased survival rate of sterile mutants on Bt247 and Bt679 is independent of the effect of *elt-2*. The data represented in the heatmap and statistics as in Fig 3E. In (A-B) worms were grown on RNAi *E. coli* HT115 plates.
(TIF)

**S6 Fig. Nuclear hormone receptor-encoding gene *nhr-8* contributes to general responses to Bt. Related to Fig 7.** Difference in survival between the N2 control and the nuclear hormone receptor mutants *nhr-8*(*ok186*). A GLM of the binomial family was fitted followed by a Tukey HSD Test (see methods section), where mutant or knockdown worm strains were compared to control strains. Asterisks show significant differences between knockout/knockdown treatment and wildtype N2 control. p-value Bonferroni adjusted. Data represented in heatmap and statistics as in Fig 3E. Triangles indicate significant difference compared to *elt-2*(RNAi).
(TIF)

**S1 Table. Differential gene expression analysis of *C. elegans* N2 infected with *B. thuringiensis* Bt247 and Bt679.**
(XLSX)

**S2 Table. Results of functional enrichment analysis using DAVID per differential gene expression cluster of *C. elegans* N2 infected with *B. thuringiensis* Bt247 and Bt679.**
(XLSX)

**S3 Table. Results of functional enrichment analysis using WormExp per differential gene expression cluster of *C. elegans* N2 infected with *B. thuringiensis* Bt247 and Bt679.**
(XLSX)

**S4 Table. Differential gene expression analysis of *C. elegans* N2 *elt-2*(RNAi) worms infected with *B. thuringiensis* Bt247 and Bt679.**
(XLSX)

**S5 Table. Results of functional enrichment analysis using DAVID per differential gene expression cluster of *C. elegans* N2 *elt-2*(RNAi) worms infected with *B. thuringiensis* Bt247 and Bt679.**
(XLSX)

**S6 Table. List of worm and bacterial strains used in this study.**
(XLSX)

**S7 Table. List of oligonucleotides used in this study.**
(XLSX)

**S8 Table. Raw phenotypic data.**
(XLSX)

**S9 Table. Results of statistical tests performed on phenotypic datasets.**
(XLSX)

## Acknowledgments

We thank Camilo Barbosa, Sabrina Butze, Li Fan, Anke Kloock, Patrick Martin, Luis Ospina, Andrei Papkou, Lena Peters, Carola Petersen, Nadja Schmitz, and the Schulenburg group for support and advice. We thank Michael Shapira and Jonathan Ewbank for valuable comments on the manuscript. We also thank James McGhee for providing the anti-ELT-2 monoclonal antibody, Michael Hengartner for the BJ49 strain, Matthias Leippe for the VC3152 strain, Christina Nielsen-LeRoux for Bt407, and Luca Signor for the MALDI-ToF analysis using the platforms of the Grenoble Instruct-ERIC Center (ISBG: UMS 3518 CNRS-CEA-UGA-EMBL) with support from FRISBI (ANR-10-INSB-05-02) and GRAL (ANR-10-LABX-49-01) within the Grenoble Partnership for Structural Biology (PSB). Knockout strains were provided either by the CGC, which is funded by NIH Office of Research Infrastructure Programs (P40 OD010440), or the National Bioresource Project coordinated by S. Mitani.

## Author Contributions

**Conceptualization:** Alejandra Zárate-Potes, Hinrich Schulenburg, Katja Dierking.

**Data curation:** Alejandra Zárate-Potes, Wentao Yang, Katja Dierking.

**Formal analysis:** Alejandra Zárate-Potes, Wentao Yang, Guillaume Tetreau, Hinrich Schulenburg, Katja Dierking.

**Funding acquisition:** Jacques-Philippe Colletier, Hinrich Schulenburg, Katja Dierking.

**Investigation:** Alejandra Zárate-Potes, Barbara Pees, Rebecca Schalkowski, Philipp Segler, Bentje Andresen, Daniela Haase, Rania Nakad, Guillaume Tetreau, Jacques-Philippe Colletier, Katja Dierking.

**Methodology:** Alejandra Zárate-Potes, Wentao Yang, Barbara Pees, Daniela Haase, Guillaume Tetreau, Hinrich Schulenburg, Katja Dierking.

**Project administration:** Alejandra Zárate-Potes, Hinrich Schulenburg, Katja Dierking.

**Resources:** Philip Rosenstiel, Jacques-Philippe Colletier, Hinrich Schulenburg, Katja Dierking.

**Supervision:** Alejandra Zárate-Potes, Hinrich Schulenburg, Katja Dierking.

**Visualization:** Alejandra Zárate-Potes, Hinrich Schulenburg, Katja Dierking.

**Writing – original draft:** Alejandra Zárate-Potes, Hinrich Schulenburg, Katja Dierking.

**Writing – review & editing:** Alejandra Zárate-Potes, Wentao Yang, Barbara Pees, Rebecca Schalkowski, Philipp Segler, Bentje Andresen, Daniela Haase, Rania Nakad, Philip Rosenstiel, Guillaume Tetreau, Jacques-Philippe Colletier, Hinrich Schulenburg, Katja Dierking.

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
