## [Decision Letter · Decision Letter 0]

27 May 2020

Dear Dr. Dierking,

Thank you very much for submitting your manuscript "A single C. elegans master regulator causes opposite infection outcomes with different strains of the same pathogen species" for consideration at PLOS Pathogens. As with all papers reviewed by the journal, your manuscript was reviewed by members of the editorial board and by several independent reviewers. In light of the reviews (below this email), we would like to invite the resubmission of a significantly-revised version that takes into account the reviewers' comments.  In particular, it is necessary that you  revise your conclusions to only reflect what can reasonably be concluded based on the data presented.

We cannot make any decision about publication until we have seen the revised manuscript and your response to the reviewers' comments. Your revised manuscript is also likely to be sent to reviewers for further evaluation.

Sincerely,

James J Collins III

Associate Editor

PLOS Pathogens

Michael Wessels

Section Editor

PLOS Pathogens

Kasturi Haldar

Editor-in-Chief

PLOS Pathogens

orcid.org/0000-0001-5065-158X

Michael Malim

Editor-in-Chief

PLOS Pathogens

orcid.org/0000-0002-7699-2064

Reviewer's Responses to Questions

**Part I - Summary**

Reviewer #1: Zárate-Potes, et al. report a significant finding that differential host response to pathogen infection is dramatically influenced by gene expression responses to that infection. This finding is unexpected because as one might predict that loss of a key transcription factor required for normal host response would only render the host compromised for all pathogens. Rather, the authors report that reduction of elt-2 by RNAi renders C. elegans more resistant to one isolate and more sensitive to another. One particularly exciting finding is shown in Figure 4 where there was a change in intestinal integrity in response to different Bt isolates despite equivalent pathogen load. Although the mechanistic basis for this differential outcome remains unclear, the outcome is well documented and will certainly be valuable to the field and spark further studies.

Reviewer #2: This manuscript describes an extensive investigation of the contrasting effects of two Bacillus thuringiensis strains (Bt679 and Bt247) on the nematode Caenorhabditis elegans. Both are able to kill C. elegans, mainly or completely by intestinal damage caused by their different Cry toxins, but the authors show by transcriptome analyses that the responses of the worms have major differences. Exposure of worms to a toxin-negative Bt strain (Bt407) has similar transcriptomic consequences to exposure to standard E. coli food, which is reassuring, but the two toxic Bt strains elicit responses that indicate both shared and distinct consequences to the different strains. The importance of these effects was explored with appropriate mutants or RNAi knockdown. Major pathways implicated in the shared responses include the AP1 factor jun-1; the p38 MAP kinase pathway contributes to resistance to Bt679 but not to Bt247; and the ELT-2 transcription factor has opposite effects in the responses to the two strains.

The most striking result is that RNAi knockdown of the major intestinal transcription factor ELT-2 (a GATA factor) caused increased sensitivity to Bt679, as might be expected, but dramatically increased survival in the presence of Bt247. The authors show that the improved survival is probably due to increased tolerance, as opposed to resistance, because the intestinal damage and pathogen loads are apparently unchanged by elt-2(RNAi).

Much of the manuscript is devoted to attempts, mostly inconclusive, to explain this remarkable effect. The authors write, “increase [of elt-2(RNA worms] in survival on Bt247 is multifactorial, influenced by the nuclear hormone receptors NHR-99 and NHR-193, and also involves lipid metabolism, detoxification, and trehalose catabolism”, but most of this involvement is guesswork and speculation, unfortunately. The putative transcription factors NHR-99 and NHR-193 are shown to have an effect, but their targets are unknown. Analysing the effects of ELT-2 is bound to be hugely complicated since it has transcriptional effects on at least one third of the C. elegans genome (e.g. page 28: 6180 genes are affected by elt-2(RNAi). Moreover, contributions of related/redundant GATA factors such as ELT-7 need to be considered. Sterility effects are not worth reporting in such detail, or including in Figure 8 unless qualified with yet more question marks.

The authors present some excellent EM data, and do a nice job of analyzing the positive contributions of ELT-2 and the p38 MAP kinase pathway to resistance.

**Part II – Major Issues: Key Experiments Required for Acceptance**

Reviewer #1: 1. The major concern with this manuscript is the authors’ concluding model that depicts a scenario where elt-2 mediates two distinct responses to different pore-forming toxin inputs. Is it not more likely that intact elt-2 mediates one response program and loss of elt-2 function shows different outcomes depending on the insult? In other words, the different outcome is the result of how loss of elt-2 alters gene expression such that it primes animals beneficially for one toxin and renders animals susceptible to the other. The authors provide some evidence for their conclusion with the nhr versus pmk-1 results but this level of epistasis analysis is not conclusive enough to depict the two responses as entirely distinct as shown in the model. If the authors’ model is correct, it would be incumbent to demonstrate that elt-2 acts not only as a mediator of transcriptional programs in response to pathogens but also as a sensor of the specific pathogen.

2. Perhaps the weakest conclusion of this study is whether necrotic cell death is or is not involved in the pathogenesis. The authors make some contradictory statements about the role of necrosis. In the text, they state they did not find consistent responses on the two Bt isolates with loss of different factors in the necrotic pathway yet in the figure legend title they state that specific responses do not involve the necrosis pathway. I think this issue needs to be clarified. Are there any cellular morphological changes in response to the two distinct Bt isolates indicating similar or different necrotic cell death outcomes?

Reviewer #2: 1. A general criticism of this work is that most of what has been looked at seems to be the consequences of intoxication rather than infection. This is especially true for Bt247, which does not appear to proliferate in the C. elegans intestine. This issue and its implications need to be better addressed.

2. The authors that purified toxins from Bt679 can kill, and showed that Cry21Aa3 rather than Cry14Aa2 is responsible, but purified Cry6Ba from Bt247 had no effect, although deletion of the relevant gene from Bt247 removed toxicity. This is disappointing and needs more explanation or investigation.

3. Most of the experimentation on p.25-28 and in Figure 6 should be greatly shortened or omitted, as much of it is negative, inconclusive or inconsistent (as the authors admit).

**Part III – Minor Issues: Editorial and Data Presentation Modifications**

Reviewer #1: 1. Figure 1 has a couple of issues that need to be amended. First, the worm image appears to be very similar to WormAtlas and that should be cited if true. Second, the legend needs to be clarified to match what is being shown in A versus B. Third, the authors indicate that microvilli are artificially colored. Anatomically, it appears that the apical membrane is what has been marked throughout. Please clarify these issues in the legend.

2. The title comes across as a bit ostentatious and uninformative. The data seem to show that elt-2 GATA factor regulates a transcriptional response that results in distinct outcomes depending on the pore toxin insult. The title should more adequately reflect those findings.

3. A major conclusion of the authors is that the jun-1 transcription factor regulates the common response to the Bt strains whereas the elt-2 transcription factor mediates the strain-specific response. One could argue that the authors have found two transcriptional response pathways acting in parallel or the elt-2 is a subset of the jun-1. Thus it is unnecessary to distinguish one as regulatory and the other as a mediator given the evidence in hand. “Mediate” could be concluded for both without diminishing the impact of the finding.

Reviewer #2: 4. Page 15 Figure 3G is either unnecessary or wrong.

5. p.22 line -7 ‘after Bt247 infection’ should be presumably be ‘after Bt679 infection’. Also, the Bt247 data in Figure 5A have triangles indicating a stronger effect in elt-2(RNAi); pmk-1 than in elt-2(RNAi) alone, but the heatmaps look identical.

6. p.32 line -6 NHR-93 should be NHR-193.

7. p. 30, 31 and throughout: Transcriptional reporters are preferably written as nhr-193p::GFP, not pnhr-193::GFP

8. p. 41 Zeiss Zeiss

PLOS authors have the option to publish the peer review history of their article (what does this mean?). If published, this will include your full peer review and any attached files.

Reviewer #1: No

Reviewer #2: No
---

## [Decision Letter · Decision Letter 1]

21 Jul 2020

Dear Dr. Dierking,

We are pleased to inform you that your manuscript 'The C. elegans GATA transcription factor elt-2 mediates distinct transcriptional responses and opposite infection outcomes towards different Bacillus thuringiensis strains' has been provisionally accepted for publication in PLOS Pathogens.

Best regards,

James J Collins III

Section Editor

PLOS Pathogens

Michael Wessels

Section Editor

PLOS Pathogens

Kasturi Haldar

Editor-in-Chief

PLOS Pathogens

orcid.org/0000-0001-5065-158X

Michael Malim

Editor-in-Chief

PLOS Pathogens

orcid.org/0000-0002-7699-2064

Reviewer Comments (if any, and for reference):

Reviewer's Responses to Questions

**Part I - Summary**

Reviewer #1: In this substantially revised and improved manuscript, Zarate-Potes, et al. have now clarified their message and strengthened their conclusions. They have worked hard to address all of my concerns including disambiguation of their summary model. Their major conclusion and exciting core finding for this report is the different responses to distinct strains of a bacterial species including a balance (perhaps tradeoff) between tolerance and resistance via different elt-2-dependent gene expression program subsets. Although the molecular mechanisms of sensing and the tradeoff of tolerance and resistance are not yet understood, these remain exciting lines of inquiry for future reports.

Reviewer #2: This is an excellent revision to an interesting and valuable piece of research. The authors' responses to criticisms are thorough and satisfactory, and the manuscript is much improved.

**Part II – Major Issues: Key Experiments Required for Acceptance**

Reviewer #1: I have no major issues with this revision.

Reviewer #2: No further issues.

**Part III – Minor Issues: Editorial and Data Presentation Modifications**

Reviewer #1: (No Response)

Reviewer #2: No further issues.

PLOS authors have the option to publish the peer review history of their article (what does this mean?). If published, this will include your full peer review and any attached files.

Reviewer #1: No

Reviewer #2: No

---

## [Editor Report · Acceptance letter]

20 Aug 2020

Dear Dr. Dierking,

We are delighted to inform you that your manuscript, "The *C. elegans* GATA transcription factor *elt-2* mediates distinct transcriptional responses and opposite infection outcomes towards different *Bacillus thuringiensis* strains," has been formally accepted for publication in PLOS Pathogens.

Best regards,

Kasturi Haldar

Editor-in-Chief

PLOS Pathogens

orcid.org/0000-0001-5065-158X

Michael Malim

Editor-in-Chief

PLOS Pathogens

orcid.org/0000-0002-7699-2064